

# Hierarchical generalized additive models in ecology: an introduction with mgcv

Eric J. Pedersen[1,2], David L. Miller[3,4], Gavin L. Simpson[5,6] and Noam Ross[7]

[1] Northwest Atlantic Fisheries Center, Fisheries and Oceans Canada, St. John's, NL, Canada
[2] Department of Biology, Memorial University of Newfoundland, St. John's, NL, Canada
[3] Centre for Research into Ecological and Environmental Modelling, University of St Andrews, St Andrews, UK
[4] School of Mathematics and Statistics, University of St Andrews, St Andrews, Scotland, UK
[5] Institute of Environmental Change and Society, University of Regina, Regina, SK, Canada
[6] Department of Biology, University of Regina, Regina, SK, Canada
[7] EcoHealth Alliance, New York, NY, USA

## ABSTRACT

In this paper, we discuss an extension to two popular approaches to modeling complex structures in ecological data: the generalized additive model (GAM) and the hierarchical model (HGLM). The hierarchical GAM (HGAM), allows modeling of nonlinear functional relationships between covariates and outcomes where the shape of the function itself varies between different grouping levels. We describe the theoretical connection between HGAMs, HGLMs, and GAMs, explain how to model different assumptions about the degree of intergroup variability in functional response, and show how HGAMs can be readily fitted using existing GAM software, the **mgcv** package in R. We also discuss computational and statistical issues with fitting these models, and demonstrate how to fit HGAMs on example data. All code and data used to generate this paper are available at: github.com/eric-pedersen/mixed-effect-gams.

## INTRODUCTION

Two of the most popular and powerful modeling techniques currently in use by ecologists are generalized additive models (GAMs; *Wood, 2017a*) for modeling flexible regression functions, and generalized linear mixed models ("hierarchical generalized linear models" (HGLMs) or simply "hierarchical models"; *Bolker et al., 2009*; *Gelman et al., 2013*) for modeling between-group variability in regression relationships.

At first glance, GAMs and HGLMs are very different tools used to solve different problems. GAMs are used to estimate smooth functional relationships between predictor variables and the response. HGLMs, on the other hand, are used to estimate linear relationships between predictor variables and response (although nonlinear relationships can also be modeled through quadratic terms or other transformations of the predictor variables), but impose a structure where predictors are organized into groups (often

Corresponding author
Eric J. Pedersen,
eric.j.pedersen@gmail.com

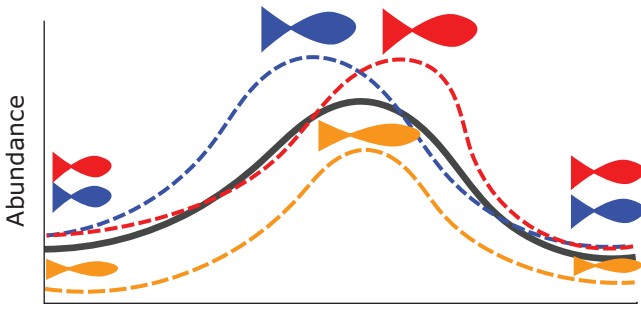

**Figure 1 Hypothetical example of functional variability between different group levels.** Each dashed line indicates how the abundance for different species of fish in a community might vary as a function of average water temperature. The orange species shows lower abundance at all temperatures, and the red and blue species differ at which temperature they can achieve the maximum possible size. However, all three curves are similarly smooth and peak close to one another relative to the entire range of tested temperatures. The solid black line represents an "average abundance curve," representing the mean abundance across species in the sample.

referred to as "blocks") and the relationships between predictor and response may vary across groups. Either the slope or intercept, or both, may be subject to grouping. A typical example of HGLM use might be to include site-specific effects in a model of population counts, or to model individual level heterogeneity in a study with repeated observations of multiple individuals.

However, the connection between HGLMs and GAMs is quite deep, both conceptually and mathematically (*Verbyla et al., 1999*). HGLMs and GAMs fit highly variable models by "pooling" parameter estimates toward one another, by penalizing squared deviations from some simpler model. In an HGLM, this occurs as group-level effects are pulled toward global effects (penalizing the squared differences between each group-level parameter estimate and the global effect). In a GAM, this occurs via the enforcement of a smoothness criterion on the variability of a functional relationship, pulling parameters toward some function that is assumed to be totally smooth (such as a straight line) by penalizing squared deviations from that totally smooth function.

Given this connection, a natural extension to the standard GAM framework is to allow smooth functional relationships between predictor and response to vary between groups, but in such a way that the different functions are in some sense pooled toward a common shape. We often want to know both how functional relationships vary between groups, and if a relationship holds across groups. We will refer to this type of model as a hierarchical GAM (HGAM).

There are many potential uses for HGAMs. For example, we can use them to estimate how the maximum size of different fish species varies along a common temperature gradient (Fig. 1). Each species will typically have its own response function, but since the species overlap in range, they should have similar responses over at least some of the temperature gradient; Fig. 1 shows all three species reach their largest maximum sizes in the center of the temperature gradient. Estimating a separate function for each species throws away a lot of shared information and could result in highly noisy function estimates

if there were only a few data points for each species. Estimating a single average relationship could result in a function that did not predict any specific group well. In our example, using a single global temperature-size relationship (Fig. 1, solid line) would miss that the three species have distinct temperature optima, and that the orange species is significantly smaller at all temperatures than the other two (Fig. 1). We prefer a hierarchical model that includes a global temperature-size curve plus species-specific curves that were penalized to be close to the mean function.

This paper discusses several approaches to group-level smoothing, and corresponding trade-offs. We focus on fitting HGAMs with the popular **mgcv** package (*Wood, 2011*) for the R statistical programming language (*R Development Core Team, 2018*), which allows for a variety of HGAM model structures and fitting strategies. We discuss options available to the modeller and practical and theoretical reasons for choosing them. We demonstrate the different approaches across a range of case studies.

This paper is divided into five sections. Part II is a brief review of how GAMs work and their relation to hierarchical models. In part III, we discuss different HGAM formulations, what assumptions each model makes about how information is shared between groups, and the different ways of specifying these models in **mgcv**. In part IV, we work through example analyses using this approach, to demonstrate the modeling process and how HGAMs can be incorporated into the ecologist's quantitative toolbox. Finally, in part V, we discuss some of the computational and statistical issues involved in fitting HGAMs in **mgcv**. We have also included all the code needed to reproduce the results in this manuscript in Supplemental Code (online), and on the GitHub repository associated with this paper: github.com/eric-pedersen/mixed-effect-gams.

## A REVIEW OF GENERALIZED ADDITIVE MODELS

The generalized linear model (GLM; *McCullagh & Nelder, 1989*) relates the mean of a response ($y$) to a linear combination of explanatory variables. The response is assumed to be conditionally distributed according to some exponential family distribution (e.g., binomial, Poisson or Gamma distributions for trial, count or strictly positive real responses, respectively). The GAM (*Hastie & Tibshirani, 1990*; *Ruppert, Wand & Carroll, 2003*; *Wood, 2017a*) allows the relationships between the explanatory variables (henceforth covariates) and the response to be described by smooth curves (usually *splines* (*De Boor, 1978*), but potentially other structures). In general, we have models of the form:

$$\mathbb{E}(Y) = g^{-1}\left(\beta_0 + \sum_{j=1}^{J} f_j(x_j)\right),$$

where $\mathbb{E}(Y)$ is the expected value of the response $Y$ (with an appropriate distribution and link function $g$), $f_j$ is a smooth function of the covariate $x_j$, $\beta_0$ is an intercept term, and $g^{-1}$ is the inverse link function. Hereafter, we will refer to these smooth functions as *smoothers*. In the example equation above, there are $j$ smoothers and each is a function of only one covariate, though it is possible to construct smoothers of multiple variables.

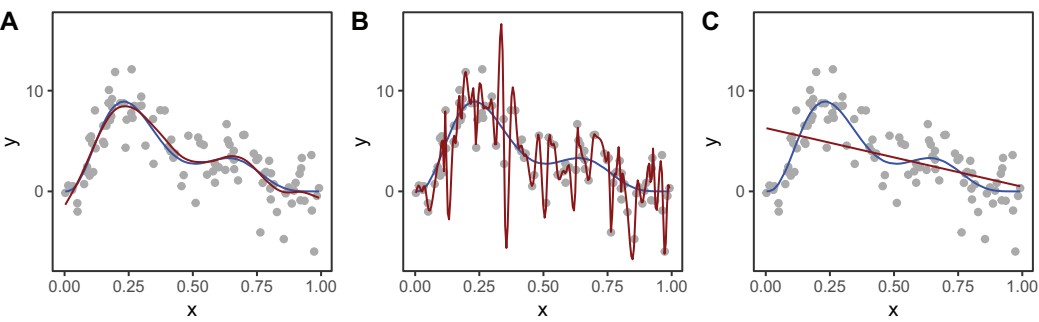

**Figure 2 Effect of different choices of smoothing parameter (λ) on the shape of the resulting smoother (red lines).** (A) λ estimated using REML; (B) λ set to zero (no smoothing); (C) λ is set to a very large value. The blue line in each panel is the known model used to simulate the data.

Each smoother $f_j$ is represented by a sum of $K$ simpler, fixed *basis functions* ($b_{j,k}$) multiplied by corresponding coefficients ($\beta_{j,k}$), which need to be estimated:

$$f_j(x_j) = \sum_{k=1}^{K} \beta_{j,k} b_{j,k}(x_j).$$

$K$, referred to as "basis size," "basis complexity," or "basis richness," determines the maximum complexity of each smoother.

It would seem that large basis size could lead to overfitting, but this is counteracted by a *smoothing penalty* that influences basis function coefficients so as to prevent excess wiggliness and ensure appropriate complexity of each smoother. For each smoother, one (or more) *penalty matrices* (**S**), specific to the form of the basis functions, is pre- and post-multiplied by the parameter vector β to calculate the penalty ($\beta^{T}\mathbf{S}\beta$). A penalty term is then subtracted from the model log-likelihood $L$, controlling the trade-off via a *smoothing parameter* (λ). The penalized log-likelihood used to fit the model is thus:

$$L - \lambda\beta^{T}\mathbf{S}\beta$$

Figure 2 shows an example of how different choices of the smoothing parameter (λ) affect the shape of the resulting smoother. Data (points) were generated from the blue function and noise added to them. In Fig. 2A, λ was selected using restricted maximum likelihood (REML) to give a good fit to the data. In Fig. 2B, λ was set to zero so the penalty has no effect and the function interpolates the data. Figure 2C shows when λ is set to a very large value, so the penalty removes all terms that have any wiggliness, giving a straight line.

To measure the complexity of a penalized smooth terms we use the *effective degrees of freedom* (EDF), which at a maximum is the number of coefficients to be estimated in the model, minus any constraints. The EDF can take noninteger values and larger values indicate more wiggly terms (see *Wood (2017a*, Section 6.1.2*)* for further details). The number of basis functions, $K$, sets a maximum for the EDF, as a smoother cannot have more than $K$ EDF. When the EDF is well below $K$, increasing $K$ generally has very little effect on the shape of the function. In general, $K$ should be set large enough to allow

for potential variation in the smoother while still staying low enough to keep computation time low (see section "Computational and Statistical Issues When Fitting HGAMs" for more on this). In **mgcv**, the function `mgcv::gam.check` can be used to determine if $K$ has been set too low.

Random effects are also "smooths" in this framework. In this case, the penalty matrix is the inverse of the correlation matrix of the basis function coefficients (*Kimeldorf & Wahba, 1970*; *Wood, 2017a*). For a simple single-level random effect to account for variation in group means (intercepts) there will be one basis function for each level of the grouping variable. The basis function takes a value of 1 for any observation in that group and 0 for any observation not in the group. The penalty matrix for these terms is a $g$ by $g$ identity matrix, where $g$ is the number of groups. This means that each group-level coefficient will be penalized in proportion to its squared deviation from zero. This is equivalent to how random effects are estimated in standard mixed effect models. The penalty term is proportional to the inverse of the variance of the fixed effect estimated by standard hierarchical model software (*Verbyla et al., 1999*).

This connection between random effects and splines extends beyond the varying-intercept case. Any single-penalty basis-function representation of a smooth can be transformed so that it can be represented as a combination of a random effect with an associated variance, and possibly one or more fixed effects. See *Verbyla et al. (1999)* or *Wood, Scheipl & Faraway (2013)* for a more detailed discussion on the connections between these approaches.

## Basis types and penalty matrices

The range of smoothers are useful for contrasting needs and have different associated penalty matrices for their basis function coefficients. In the examples in this paper, we will use three types of smoothers: thin plate regression splines (TPRS), cyclic cubic regression splines (CRS), and random effects.

Thin plate regression splines (*Wood, 2003*) are a general purpose spline basis which can be used for problems in any number of dimensions, provided one can assume that the amount of smoothing in any of the covariates is the same (so called isotropy or rotational invariance). TPRS, like many splines, use a penalty matrix made up of terms based on the integral of the squared derivatives of basis functions across their range (see *Wood (2017a)* page 216 for details on this penalty). Models that overfit the data will tend to have large derivatives, so this penalization reduces wiggliness. We will refer to the order of penalized derivatives by $m$. Typically, TPRS are second-order ($m = 2$), meaning that the penalty is proportionate to the integral of the squared second derivative. However, TPRS may be of lower order ($m = 1$, penalizing squared first derivatives), or higher order ($m > 2$, penalizing squared higher order derivatives). We will see in section "What are Hierarchical GAMs?" how lower-order TPRS smoothers are useful in fitting HGAMs. Example basis functions and penalty matrix **S** for a $m = 2$ TPRS with six basis functions for evenly spaced data are shown in Fig. 3.

Cyclic cubic regression splines are another smoother that penalizes the squared second derivative of the smooth across the function. In cyclic CRS the start and end of the

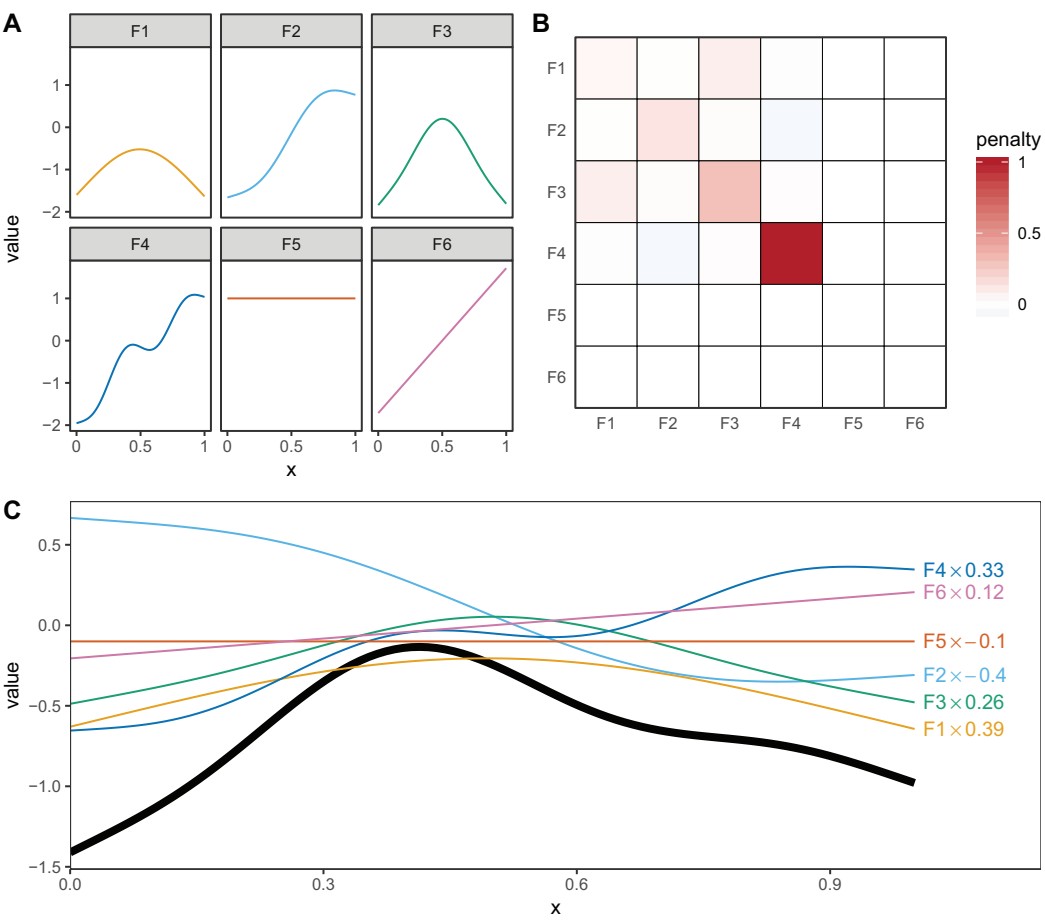

**Figure 3** (A) Examples of the basis functions associated with a six basis function thin plate regression spline (TPRS, *m* = 2), calculated for data, *x*, spread evenly between *x* = 0 and *x* = 1. Each line represents a single basis function. (B) The smoothing penalty matrix for the thin plate smoother. Red entries indicate positive values and blue indicate negative values. For example, functions F3 and F4 would have the greatest proportionate effect on the total penalty (as they have the largest values on the diagonal), whereas function F5 and F6 would not contribute to the wiggliness penalty at all (all the values in the fifth and sixth row and column of the penalty matrix are zero). This means these functions are in the null space of the penalty matrix, and are treated as completely smooth. (C) An example of how the basis functions add up to create a single smooth function. Thin colored lines represent each basis function multiplied by a coefficient, and the solid black line is the sum of those basis functions.

smoother are constrained to match in value and first derivative. These are useful for fitting models with cyclic components such as seasonal effects. We will use these smoothers to demonstrate how to fit HGAMs to cyclic data.

## Smoothing penalties vs. shrinkage penalties

Penalties can have two effects on how well a model fits: they can penalize how wiggly a given term is (smoothing) and they can penalize the absolute size of the function (shrinkage). The penalty can only affect the components of the smoother that have derivatives (the *range space*), not the other parts (the *null space*). For one-dimensional

TPRS (when $m = 2$), this means that there is a linear term (F5) left in the model, even when the penalty is in full force (as $\lambda \rightarrow \infty$), as shown in Fig. 3. (This is also why Fig. 2C shows a linear, rather than flat, fit to the data). The random effects smoother we discussed earlier is an example of a pure shrinkage penalty; it penalizes all deviations away from zero, no matter the pattern of those deviations. This will be useful later in "What are Hierarchical GAMs?," where we use random effect smoothers as one of the components of a HGAM.

## Interactions between smooth terms

It is also possible to create interactions between covariates with different smoothers (or degrees of smoothness) assumed for each covariate, using *tensor products*. For instance, if one wanted to estimate the interacting effects of temperature and time (in seconds) on some outcome, it would not make sense to use a two-dimensional TPRS smoother, as that would assume that a one degree change in temperature would equate to a 1 s change in time. Instead, a tensor product allows us to create a new set of basis functions that allow for each marginal function (here temperature and time) to have its own marginal smoothness penalty. A different basis can be used in each marginal smooth, as required for the data at hand.

There are two approaches used in **mgcv** for generating tensor products. The first approach (*Wood, 2006a*) essentially creates an interaction of each pair of basis functions for each marginal term, and a penalty for each marginal term that penalizes the average wiggliness in that term; in **mgcv**, these are created using the `te()` function. The second approach (*Wood, Scheipl & Faraway, 2013*) separates each penalty into penalized (range space) and unpenalized components (null space; components that don't have derivatives, such as intercept and linear terms in a one-dimensional cubic spline). This approach creates new basis functions and penalties for all pair-wise combinations of penalized and unpenalized components between all pairs of marginal bases; in **mgcv**, these are created using the `t2()` function. The advantage of the first method is that it requires fewer smoothing parameters, so is faster to estimate in most cases. The advantage of the second method is that the tensor products created this way only have a single penalty associated with each marginal basis (unlike the `te()` approach, where each penalty applies to all basis functions), so it can be fitted using standard mixed effect software such as **lme4** (*Bates et al., 2015*).

## Comparison to hierarchical linear models

Hierarchical generalized linear models (*Gelman, 2006*; HGLMs; also referred to as generalized linear mixed effect models, multilevel models, etc.; e.g., *Bolker et al., 2009*) are an extension of regression modeling that allows the inclusion of terms in the model that account for structure in the data—the structure is usually of the form of a nesting of the observations. For example, in an empirical study, individuals may be nested within sample sites, sites are nested within forests, and forests within provinces. The depth of the nesting is limited by the fitting procedure and number of parameters to estimate.

Hierarchical generalized linear models are a highly flexible way to think about grouping in ecological data; the groupings used in models often refer to the spatial or temporal scale of the data (*McMahon & Diez, 2007*) though can be based on any useful grouping.

We would like to be able to think about the groupings in our data in a similar way, even when the covariates in our model are related to the response in a smooth way. The next section investigates the extension of the smoothers we showed above to the case where observations are grouped and we model group-level smoothers.

## WHAT ARE HIERARCHICAL GAMS?

### What do we mean by hierarchical smoothers?

In this section, we will describe how to model intergroup variability using smooth curves and how to fit these models using **mgcv**. All models were fitted using **mgcv** version 1.8–26 (*Wood, 2011*). Model structure is key in this framework, so we start with three model choices:

1. Should each group have its own smoother, or will a common smoother suffice?
2. Do all of the group-specific smoothers have the same wiggliness, or should each group have its own smoothing parameter?
3. Will the smoothers for each group have a similar shape to one another—a shared global smoother?

These three choices result in five possible models (Fig. 4):

1. A single common smoother for all observations; We will refer to this as model *G*, as it only has a Global smoother.
2. A global smoother plus group-level smoothers that have the same wiggliness. We will refer to this as model *GS* (for Global smoother with individual effects that have a Shared penalty).
3. A global smoother plus group-level smoothers with differing wiggliness. We will refer to this as model *GI* (for Global smoother with individual effects that have Individual penalties).
4. Group-specific smoothers without a global smoother, but with all smoothers having the same wiggliness. We will refer to this as model *S*.
5. Group-specific smoothers with different wiggliness. We will refer to this as model *I*.

It is important to note that "similar wiggliness" and "similar shape" are two distinct concepts; functions can have very similar wiggliness but very different shapes. Wiggliness measures how quickly a function changes across its range, and it is easy to construct two functions that differ in shape but have the same wiggliness. For this paper, we consider two functions to have similar shape if the average squared distance between the functions is small (assuming the functions have been scaled to have a mean value of zero across their ranges). This definition is somewhat restricted; for instance, a cyclic function would not be considered to have the same shape as a phase-shifted version of the same function,

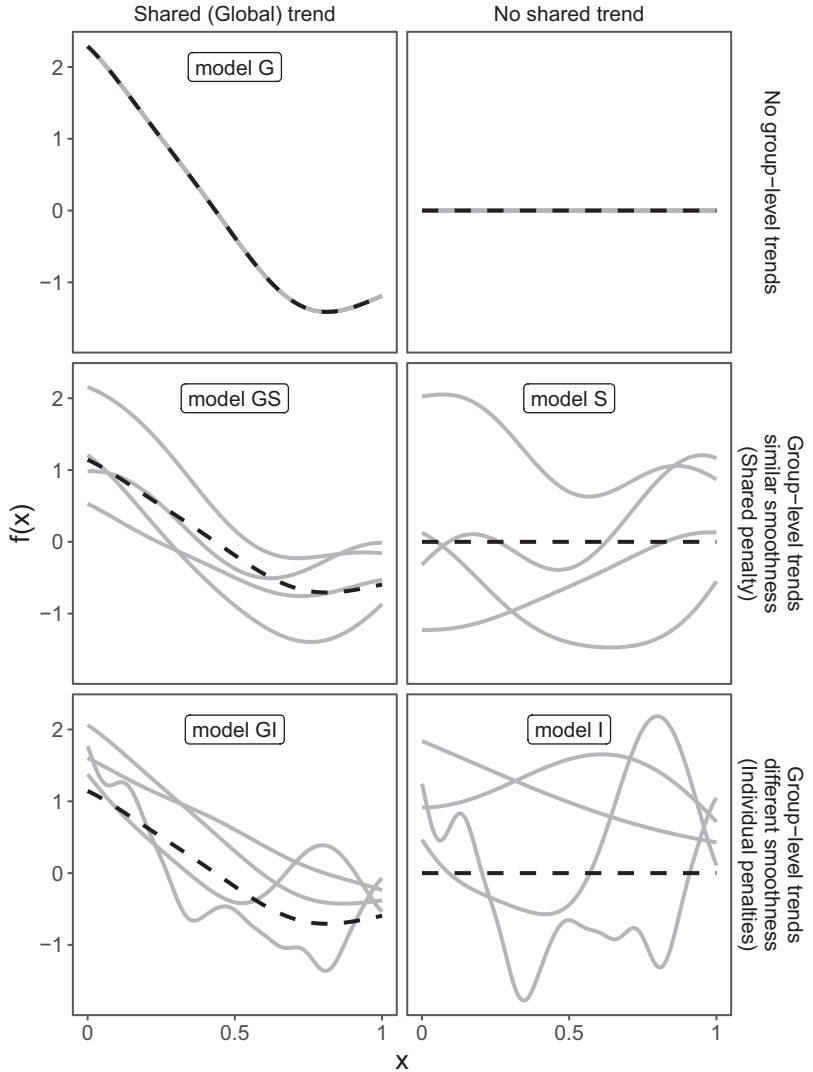

**Figure 4 Alternate types of functional variation $f(x)$ that can be fitted with HGAMs.** The dashed line indicates the average function value for all groups, and each solid line indicates the functional value at a given predictor value for an individual group level. The null model (of no functional relationship between the covariate and outcome, top right), is not explicitly assigned a model name.

nor would two normal distributions with the same mean but different standard deviations. The benefit of this definition of shape, however, is that it is straightforward to translate into penalties akin to those described in the section "A Review of Generalized Additive Models." Figure 4, model *S* illustrates the case where models have different shapes. Similarly, two curves could have very similar overall shape, but differ in their wiggliness. For instance, one function could be equal to another plus a high-frequency oscillation term. Figure 4, model *GI* illustrates this.

We will discuss the trade-offs between different models and guidelines about when each of these models is appropriate in the section "Computational and statistical issues when fitting HGAMs". The remainder of this section will focus on how to specify each of these five models using **mgcv**.

## Coding hierarchical GAMs in R

Each of the models in Fig. 4 can be coded straightforwardly in **mgcv**. We will use two example datasets to demonstrate how to code these models (see the Supplemental Code to reproduce these examples):

A. The `CO2` dataset, available in R via the **datasets** package. This data is from an experimental study by *Potvin, Lechowicz & Tardif (1990)* of $CO_2$ uptake in grasses under varying concentrations of $CO_2$, measuring how concentration-uptake functions varied between plants from two locations (Mississippi and Quebec) and two temperature treatments (chilled and warm). A total of 12 plants were used and $CO_2$ uptake measured at seven $CO_2$ concentrations for each plant (Fig. 5A). Here, we will focus on how to use HGAMs to estimate interplant variation in functional responses. This data set has been modified from the default version available with R, to recode the `Plant` variable as an unordered factor `Plant_uo`[1].

B. Data generated from a hypothetical study of bird movement along a migration corridor, sampled throughout the year (see Supplemental Code). This dataset consists of simulated sample records of numbers of observed locations of 100 tagged individuals each from six species of bird, at 10 locations along a latitudinal gradient, with one observation taken every 4 weeks. Counts were simulated randomly for each species in each location and week by creating a species-specific migration curve that gave the probability of finding an individual of a given species in a given location, then simulated the distribution of individuals across sites using a multinomial distribution, and subsampling that using a binomial distribution to simulate observation error (i.e., not every bird present at a location would be detected). The data set (`bird_move`) consists of the variables `count`, `latitude`, `week`, and `species` (Fig. 5B). This example allows us to demonstrate how to fit these models with interactions and with non-normal (count) data. The true model used to generate this data was model *GS*: a single global function plus species-specific deviations around that global function.

Throughout the examples we use REML to estimate model coefficients and smoothing parameters. We strongly recommend using either REML or marginal likelihood (ML) rather than the default generalized cross-validation criteria when fitting GAMs, for the reasons outlined in *Wood (2011)*. In each case some data processing and manipulation has been done to obtain the graphics and results below. See Supplemental Code for details on data processing steps. To illustrate plots, we will be using the `draw()` function from the **gratia** package. This package was developed by one of the authors (*Simpson, 2018*) as a set of tools to extend plotting and analysis of **mgcv** models. While **mgcv** has plotting capabilities (through `plot()` methods), **gratia** expands these by creating **ggplot2** objects (*Wickham, 2016*) that can be more easily extended and modified.

## A single common (global) smoother for all observations (Model *G*)

We start with the simplest model from the framework and include many details here to ensure that readers are comfortable with the terminology and R functions.

[1] Note that **mgcv** requires that grouping or categorical variables be coded as factors in R; it will raise an error message if passed data coded as characters. It is also important to know whether the factor is coded as ordered or unordered (see `?factor` for more details on this). This matters when fitting group-level smoothers using the `by=` argument (as is used for fitting models *GI* and *I*, shown below). If the factor is unordered, **mgcv** will set up a model with one smoother for each grouping level. If the factor is ordered, **mgcv** will set any basis functions for the first grouping level to zero. In model *GI* the ungrouped smoother will then correspond to the first grouping level, rather than the average functional response, and the group-specific smoothers will correspond to deviations from the first group. In model *I*, using an ordered factor will result in the first group not having a smoother associated with it at all.

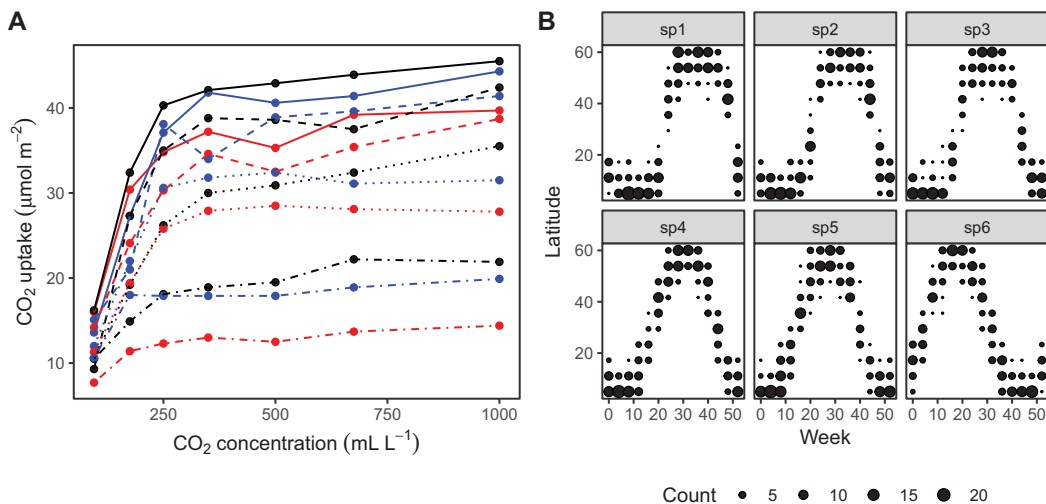

**Figure 5 Example data sets used throughout section "What are Hierarchical GAMs?"** (A) Grass $CO_2$ uptake vs. $CO_2$ concentration for 12 individual plants. Color and line type included to distinguish individual plant trends. (B) Simulated data set of bird migration, with point size corresponding to weekly counts of six species along a latitudinal gradient (zeros excluded for clarity).

For our `CO2` data set, we will model $\log_e(\texttt{uptake})$ as a function of two smoothers: a TPRS of $\log_e$-concentration, and a random effect for `Plant_uo` to model plant-specific intercepts. Mathematically:

$$\log_e(\texttt{uptake}_i) = f(\log_e(\texttt{conc}_i)) + \zeta_{\texttt{Plant\_uo}} + \varepsilon_i$$

where $\zeta_{\texttt{Plant\_uo}}$ is the random effect for plant and $\varepsilon_i$ is a Gaussian error term. Here, we assume that $\log_e(\texttt{uptake}_i)$ is normally distributed.

In R we can write our model as:

```
CO2_modG <- gam(log(uptake) ~ s(log(conc), k=5, bs="tp") +
                    s(Plant_uo, k=12, bs="re"),
                data=CO2, method="REML", family="gaussian")
```

This is a common GAM structure, with a single smooth term for each variable. Specifying the model is similar to specifying a GLM in R via `glm()`, with the addition of `s()` terms to include one-dimensional or isotropic multidimensional smoothers. The first argument to `s()` are the terms to be smoothed, the type of smoother to be used for the term is specified by the `bs` argument, and the maximum number of basis functions is specified by `k`. There are different defaults in **mgcv** for *K*, depending on the type of smoother chosen; here we use a TPRS smoother (`bs="tp"`) for the concentration smoother, and set `k=5` as there are only seven separate values of concentration measured, so the default `k=10` (for TPRS) would be too high; further, setting `k=5` saves on computational time (see section "Computational and Statistical Issues When Fitting HGAMs"). The random effect smoother (`bs="re"`) that we used for the `Plant_uo` factor always has a `k` value equal to the number of levels in the grouping variable (here, 12). We specified `k=12` just to make this connection apparent.

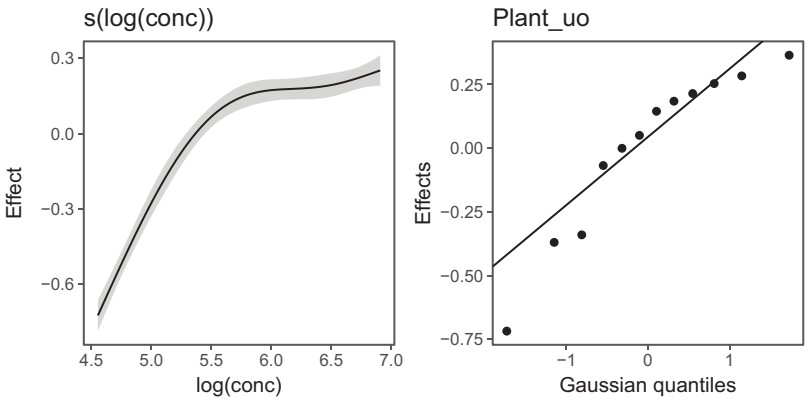

**Figure 6 gratia plotting output for model _G_ applied to the CO2 dataset.** `s(log(conc))`: the smoother of $\log_e$ concentration. `Plant_uo`: a quantile–quantile plot of the random effects against Gaussian quantiles, used to check the appropriateness of the normal random effect assumption.

Figure 6 Illustrates the output of `gratia`'s `draw()` function for `CO2_modG`: the panel labelled `s(log(conc))` shows the estimated smoother for concentration, and the panel labelled `Plant_uo` shows a quantile–quantile plot of the estimated random effects vs. Gaussian quantiles, which can be used to check our model.

Looking at the effects by term is useful, but we are often interested in fitted values or predictions from our models. Using the built in prediction functions with **mgcv**, we can estimate what the fitted function (and uncertainty around it) should look like for each level, as shown in Fig. 7 (see Supplemental Code for more details on how to generate these predictions).

Examining these plots, we see that while functional responses among plants are similar, some patterns are not captured by this model. For instance, for plant Qc2 the model clearly underestimates $CO_2$ uptake. A model including individual differences in functional responses may better explain variation.

For our bird example, we model the count of birds as a function of location and time, including their interaction. For this we structure the model as:

$$\mathbb{E}(\texttt{count}_i) = \exp(f(\texttt{week}_i, \texttt{latitude}_i))$$

where we assume that $\texttt{Count}_i \sim$ Poisson. For the smooth term, _f_, we employ a tensor product of `latitude` and `week`, using a TPRS for the marginal latitude effect, and a cyclic CRS for the marginal week effect to account for the cyclic nature of weekly effects (we expect week 1 and week 52 to have very similar values)[2], both splines had basis complexity (`k`) of 10.

```
bird_modG <- gam(count ~ te(week, latitude, bs=c("cc", "tp"), k=c(10, 10)),
                 data=bird_move, method="REML", family="poisson",
                 knots=list(week=c(0, 52)))
```

Figure 8 shows the default `draw(bird_modG)` plot for the week-by-latitude smoother. It shows birds starting at low latitudes in the winter then migrating to high latitudes from the 10th to 20th week, staying there for 15–20 weeks, then migrating back.

[2] The cyclic smoother requires that the start and end points of the cyclic variable are specified, via the `knots` argument; the smoother will have the exact same value at the start and end. In the absence of a specified start and end point, gam will assume the end points are the smallest and largest observed levels of the covariate (see `mgcv::smooth.con-struct.cc.smooth.spec` for more details). Note that in `bird_modG` we have specified week 0 and week 52 as the endpoints, as the first (week 1) and last weeks (week 52) of the year should not have exactly the same expected value.

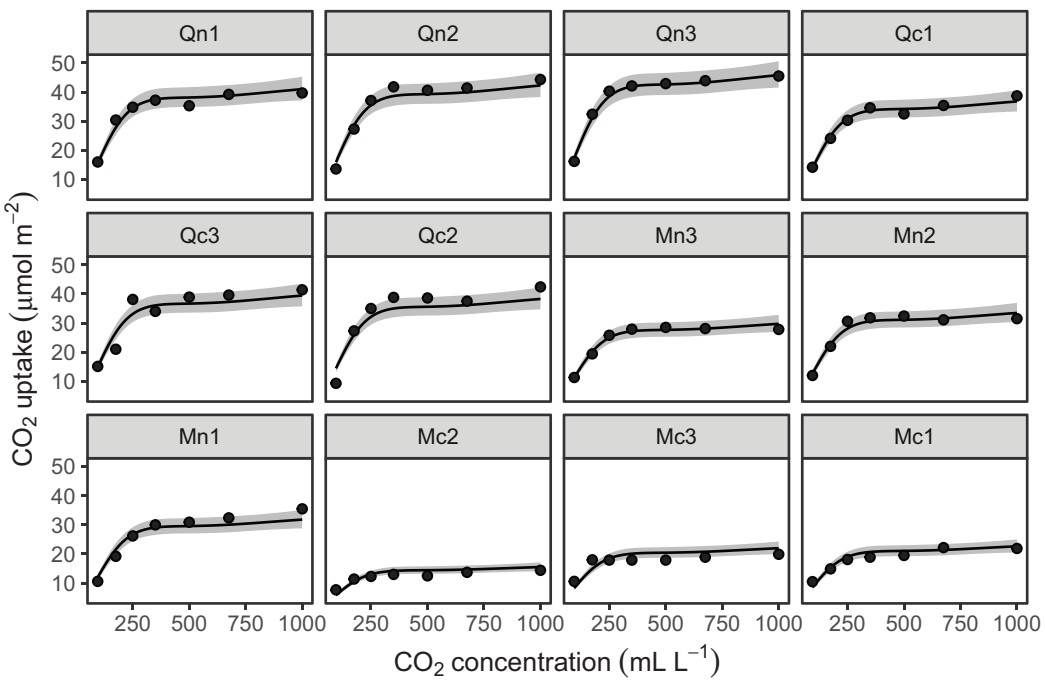

**Figure 7 Predicted uptake function (±2 s.e.) for each plant, based on model *G* (a single global function for uptake plus a individual-level random effect intercept).** Model predictions are for log-uptake, but are transformed here to show the fitted function on the original scale of the data.

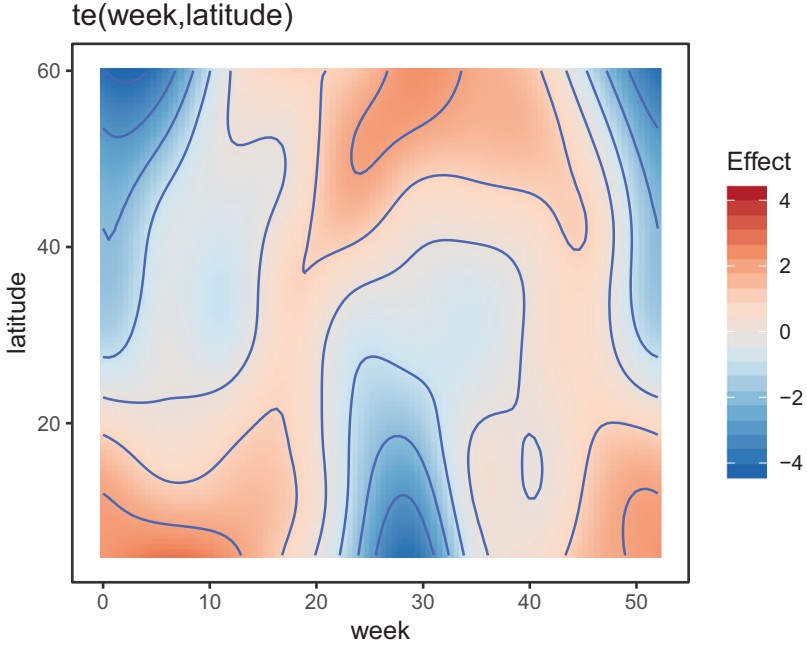

**Figure 8 Plot illustrating the average log-abundance of all bird species at each latitude for each week, with red colors indicating more individuals and blue colors fewer.**

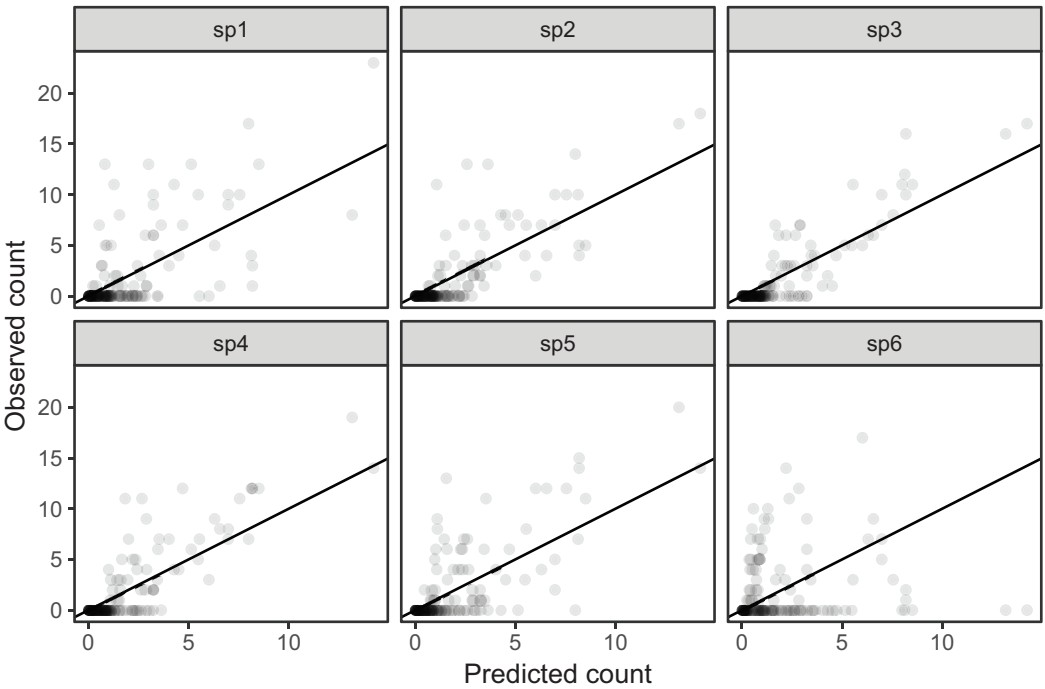

**Figure 9 Observed counts by species vs. predicted counts from `bird_modG` (1–1 line added as reference).** If our model fitted well we would expect that all species should show similar patterns of dispersion around the 1-1 line (and as we are assuming the data is Poisson, the variance around the mean should equal the mean). Instead we see that variance around the predicted value is much higher for species 1 and 6.

However, the plot also indicates a large amount of variability in the timing of migration. The source of this variability is apparent when we look at the timing of migration of each species (cf. Fig. 5B).

All six species in Fig. 5B show relatively precise migration patterns, but they differ in the timing of when they leave their winter grounds and the amount of time they spend at their summer grounds. Averaging over all of this variation results in a relatively imprecise (diffuse) estimate of migration timing (Fig. 8), and viewing species-specific plots of observed vs. predicted values (Fig. 9), it is apparent that the model fits some of the species better than others. This model could potentially be improved by adding intergroup variation in migration timing. The rest of this section will focus on how to model this type of variation.

## A single common smoother plus group-level smoothers that have the same wiggliness (model GS)

Model *GS* is a close analogue to a GLMM with varying slopes: all groups have similar functional responses, but intergroup variation in responses is allowed. This approach works by allowing each grouping level to have its own functional response, but penalizing functions that are too far from the average.

This can be coded in **mgcv** by explicitly specifying one term for the global smoother (as in model *G* above) then adding a second smooth term specifying the group-level smooth terms,

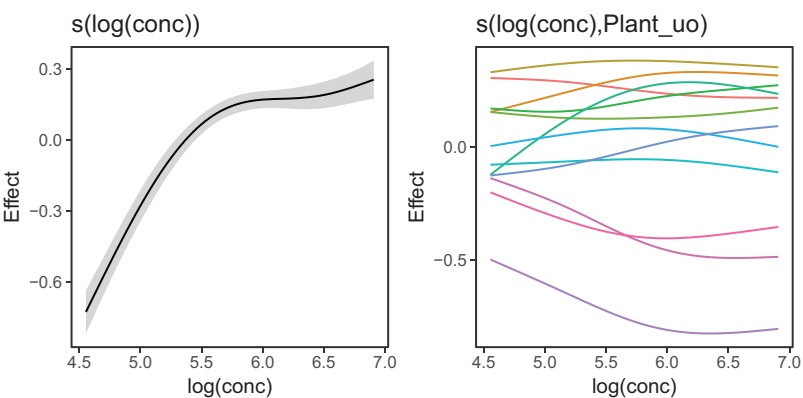

**Figure 10 Global function (`s(log(conc))`) and group-specific deviations from the global function (`s(log(conc),Plant_uo)`) for `CO2_modGS`.**

using a penalty term that tends to draw these group-level smoothers toward zero. **mgcv** provides an explicit basis type to do this, the factor-smoother interaction or `"fs"` basis (see `?mgcv::factor.smooth.interaction` for details). This smoother creates a copy of each set of basis functions for each level of the grouping variable, but only estimates one smoothing parameter for all groups. To ensure that all parts of the smoother can be shrunk toward zero effect, each component of the penalty null space is given its own penalty[3].

We modify the previous $CO_2$ model to incorporate group-level smoothers as follows:

$$\log_e(\texttt{uptake}_i) = f(\log_e(\texttt{conc}_i)) + f_{\texttt{Plant\_uo}_i}(\log_e(\texttt{conc}_i)) + \varepsilon_i$$

where $f_{\texttt{Plant\_uo}_i}(\log_e(\texttt{conc}_i))$ is the smoother for concentration for the given plant. In R we then have:

```
CO2_modGS <- gam(log(uptake) ~ s(log(conc), k=5, m=2) +
                 s(log(conc), Plant_uo, k=5, bs="fs", m=2),
             data=CO2, method="REML")
```

Figure 10 shows the fitted smoothers for `CO2_modGS`. The plots of group-specific smoothers indicate that plants differ not only in average log-uptake (which would correspond to each plant having a straight line at different levels for the group-level smoother), but differ slightly in the shape of their functional responses. Figure 11 shows how the global and group-specific smoothers combine to predict uptake rates for individual plants. We see that, unlike in the single global smoother case above, none of the curves deviate from the data systematically.

The factor-smoother interaction-based approach mentioned above does not work for higher-dimensional tensor product smoothers (`fs()` does still work for higher dimensional isotropic smoothers). Instead, the group-specific term can be specified with a tensor product of the continuous smoothers and a random effect for the grouping parameter[4]. e.g.:

```
y ~ te(x1, x2, bs="tp", m=2) +
    t2(x1, x2, fac, bs=c("tp","tp","re"), m=2, full=TRUE)
```

[3] As part of the penalty construction, each group will also have its own intercept (part of the penalized null space), so there is no need to add a separate term for group specific intercepts as we did in model *G*.

[4] As mentioned in the section "A Review of Generalized Additive Models," these terms can be specified either with `te()` or `t2()` terms. Using `t2` as above (with `full=TRUE`) is essentially a multivariate equivalent of the factor-smoother interaction; it requires more smooth terms than `te()`, but can be fit using other mixed effects software such as **lme4**, which is useful when fitting models with a large number of group levels (see section "Computational and Statistical Issues When Fitting HGAMs" on computational issues for details). We have generally found that `t2(full=TRUE)` is the best approach for multidimensional *GS* models when the goal is to accurately estimate the global smoother in the presence of group-level smoothers; other approaches (using `te()`) tend to result in the global smoother being overly penalized toward the flat function, and the bulk of the variance being assigned to the group-level smoother. We discuss this further in the section "Computational and Statistical Issues When Fitting HGAMs," "Estimation issues when fitting both global and group-level smoothers."

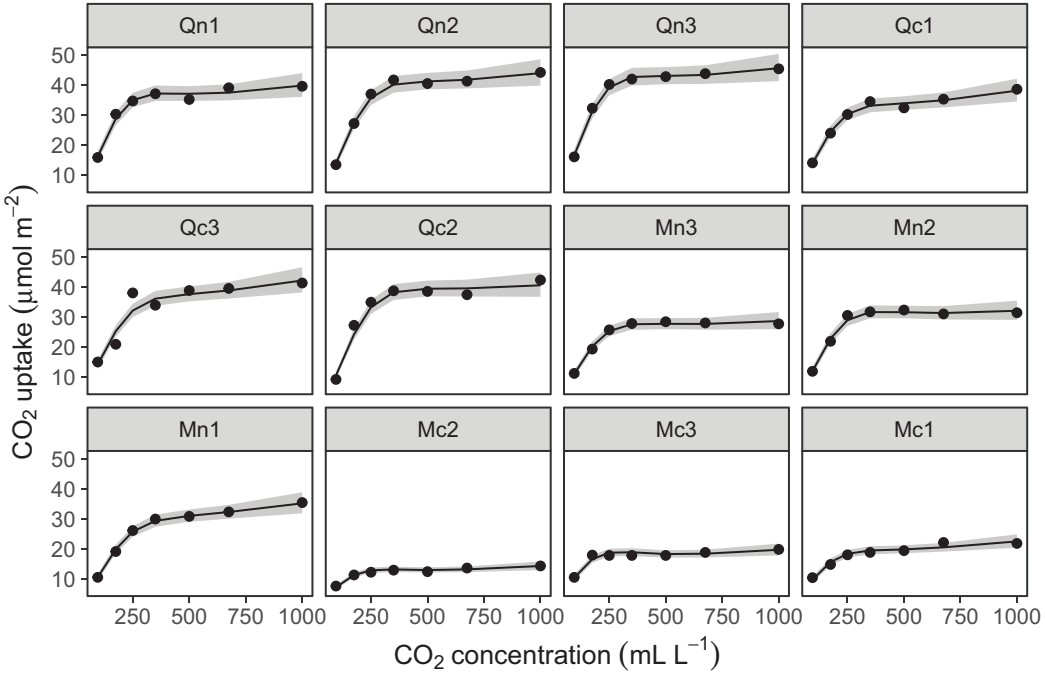

**Figure 11 Predicted uptake values (lines) vs. observed uptake for each plant, based on model _GS_.**

We illustrate this approach below on the bird migration data.

```
bird_modGS <- gam(count ~ te(week, latitude, bs=c("cc", "tp"),
                      k=c(10, 10), m=2) +
              t2(week, latitude, species, bs=c("cc", "tp", "re"),
                  k=c(10, 10, 6), m=2, full=TRUE),
              data=bird_move, method="REML", family="poisson",
              knots=list(week=c(0, 52)))
```

Model _GS_ is able to effectively capture the observed patterns of interspecific variation in migration behavior (Fig. 12A). It shows a much tighter fit between observed and predicted values, as well as less evidence of over-dispersion in some species compared to model _G_ (Fig. 12B).

## A single common smoother plus group-level smoothers with differing wiggliness (Model _GI_)

This model class is very similar to model _GS_, but we now allow each group-specific smoother to have its own smoothing parameter and hence its own level of wiggliness. This increases the computational cost of the model (as there are more smoothing parameters to estimate), and means that the only information shared between groups is through the global smoother, the common error term, and through the random effect for group-level intercepts (if used). This is useful if different groups differ substantially in how wiggly they are.

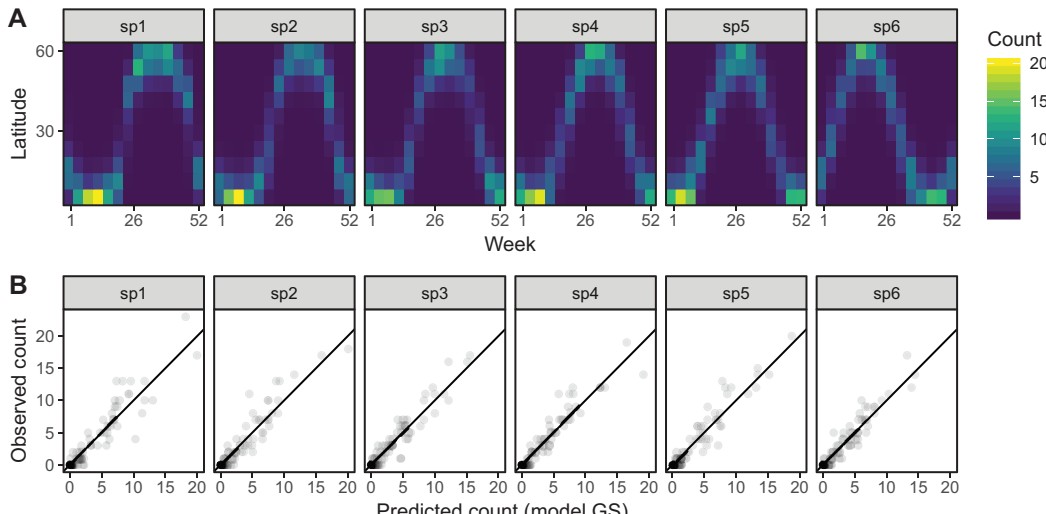

**Figure 12** (A) Predicted migration paths for each species based on `bird_modGS`, with lighter colors corresponding to higher predicted counts. (B) Observed counts vs. predictions from `bird_modGS`.

Fitting a separate smoother (with its own penalties) can be done in **mgcv** by using the `by` argument in the `s()` and `te()` (and related) functions. Therefore, we can code the formula for this model as:

```
y ~ s(x, bs="tp") + s(x, by=fac, m=1, bs="tp") + s(fac, bs="re")
```

Note two major differences here from how model *GS* was specified:

1. We explicitly include a random effect for the intercept (the `bs="re"` term), as group-specific intercepts are not incorporated into factor `by` variable smoothers (as would be the case with a factor smoother or a tensor product random effect).

2. We specify `m=1` instead of `m=2` for the group-level smoothers, which means the marginal TPRS basis for this term will penalize the squared first derivative of the function, rather than the second derivative. This, also, reduces colinearity between the global smoother and the group-specific terms which occasionally leads to high uncertainty around the global smoother (see section "Computational and Statistical Issues When Fitting HGAMs" for more details). TPRS with `m=1` have a more restricted null space than `m=2` smoothers, so should not be as collinear with the global smoother (*Wieling et al., 2016*; *Baayen et al., 2018*). We have observed that this is much more of an issue when fitting model *GI* compared to model *GS*.

We modify the CO2 model to follow this approach like so:

```
CO2_modGI <- gam(log(uptake) ~ s(log(conc), k=5, m=2, bs="tp") +
                s(log(conc), by=Plant_uo, k=5, m=1, bs="tp") +
                s(Plant_uo, bs="re", k=12),
               data=CO2, method="REML")
```

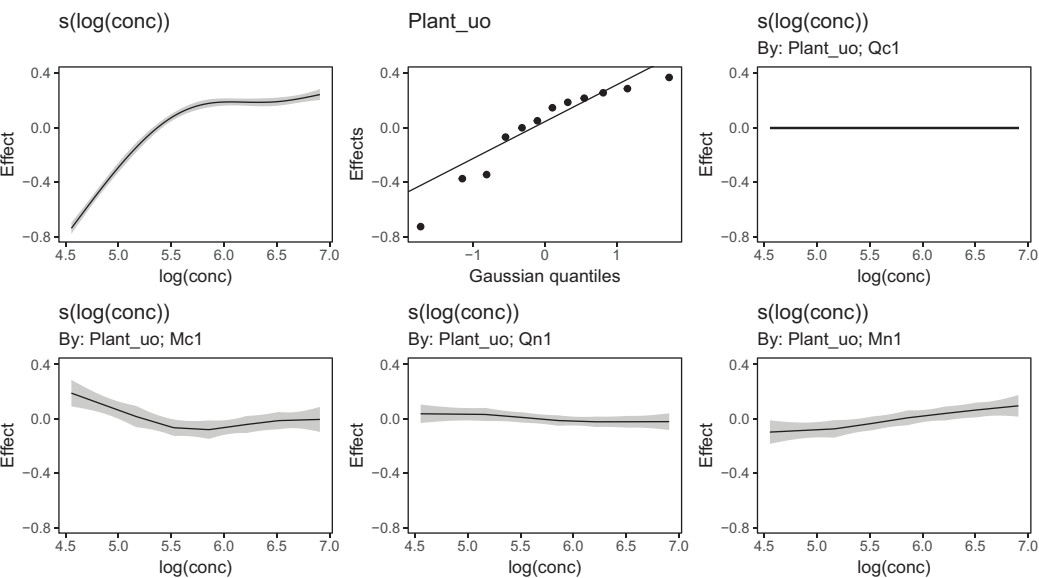

**Figure 13 Functional relationships for the CO2 data estimated for model *GI*.** s(log(conc)): the global smoother; Plant_uo: species-specific random effect intercepts. The remaining plots are a selected subset of the plant-specific smoothers, indicating how the functional response of that plant differs from the global smoother.

Figure 13 shows a subsample of the group-specific smoothers from this model. It is apparent from this that some groups (e.g., Qc1) have very similar shapes to the global smoother (differing only in intercept), others do differ from the global trend, with higher uptake at low concentrations and lower uptake at higher concentrations (e.g., Mc1, Qn1), or the reverse pattern (e.g., Mn1).

Using model *GI* with higher-dimensional data is also straightforward; by terms work just as well in tensor-product smoothers as they do with isotropic smoothers. We can see this with our bird model:

```
bird_modGI <- gam(count ~ species +
                te(week, latitude, bs=c("cc", "tp"), k=c(10, 10), m=2) +
                te(week, latitude, by=species, bs= c("cc", "tp"),
                   k=c(10, 10), m=1),
              data=bird_move, method="REML", family="poisson",
              knots=list(week=c(0, 52)))
```

As above, we have set (m=1) for the latitude marginal effect to avoid issues of collinearity between the global and group-level smoother. Note that switching m=1 to m=2 does not have any effect on the marginal basis for week, where we are using a cyclic smoother instead of a TPRS.

The fitted model for bird_modGI is visually indistinguishable from bird_modGS (Fig. 12) so we do not illustrate it here.

### Models without global smoothers (models *S* and *I*)

We can modify the above models to exclude the global term (which is generally faster; see section "Computational and Statistical Issues When Fitting HGAMs"). When we do not

model the global term, we are allowing each group to be differently shaped without restriction. Though there may be some similarities in the shape of the functions, these models' underlying assumption is that group-level smooth terms do not share or deviate from a common form.

*Model S*

Model *S* (shared smoothers) is model *GS* without the global smoother term; this type of model takes the form: `y~s(x, fac, bs="fs")` or `y~t2(x1, x2, fac, bs=c("tp", "tp", "re"))` in `mgcv`. This model assumes all groups have the same smoothness, but that the individual shapes of the smooth terms are not related. Here, we do not plot these models; the model plots are very similar to the plots for model *GS*. This will not always be the case. If in a study there are very few data points in each grouping level (relative to the strength of the functional relationship of interest), estimates from model *S* will typically be much more variable than from model *GS*; there is no way for the model to share information on function shape between grouping levels without the global smoother. See section "Computational and Statistical Issues When Fitting HGAMs" on computational issues for more on how to choose between different models.

```
CO2_modS <- gam(log(uptake) ~ s(log(conc), Plant_uo, k=5, bs="fs", m=2),
                data=CO2, method="REML")

bird_modS <- gam(count ~ t2(week, latitude, species, bs=c("cc", "tp", "re"),
                            k=c(10, 10, 6), m=2, full=TRUE),
                 data=bird_move, method="REML", family="poisson",
                 knots=list(week=c(0, 52)))
```

*Model I*

Model *I* is model *GI* without the first term: `y~fac+s(x, by=fac)` or `y~fac+te(x1,x2, by=fac)` (as above, plots are very similar to model *GI*).

```
CO2_modI <- gam(log(uptake) ~ s(log(conc), by=Plant_uo, k=5, bs="tp", m=2) +
                    s(Plant_uo, bs="re", k=12),
                data=CO2, method="REML")

bird_modI <- gam(count ~ species + te(week, latitude, by=species,
                                      bs=c("cc", "tp"), k=c(10, 10), m=2),
                 data=bird_move, method="REML", family="poisson",
                 knots=list(week=c(0, 52)))
```

## Comparing different HGAM specifications

These models can be compared using standard model comparison tools. Model *GS* and model *GI* will generally be nested in model *G* (depending on how each model is specified) so comparisons using generalized likelihood ratio tests (GLRTs) may be used to test if group-level smoothers are necessary (if fit using `method="ML"`). However, we do not currently recommend this method. There is not sufficient theory on how accurate

**Table 1  AIC table comparing model fits for example datasets.**

| Model | df | AIC | deltaAIC |
|---|---|---|---|
| A. CO2 models | | | |
| CO2_modG | 17 | −119 | 101 |
| CO2_modGS | 39 | −199 | 22 |
| CO2_modGI | 42 | −216 | 4 |
| CO2_modS | 53 | −219 | 1 |
| CO2_modI | 56 | −220 | 0 |
| B. bird_move models | | | |
| bird_modG | 51 | 3,374 | 1,823 |
| bird_modGS | 140 | 1,554 | 4 |
| bird_modGI | 208 | 1,682 | 132 |
| bird_modS | 127 | 1,550 | 0 |
| bird_modI | 200 | 1,634 | 84 |

parametric $p$-values are for comparing these models; there is uncertainty about what degrees of freedom to assign to models with varying smoothness, and slightly different model specifications may not result in nested models (See *Wood (2017a)* Section 6.12.4 and `?mgcv::anova.gam` for more discussion on using GLRTs to compare GAMs).

Comparing models based on AIC is a more robust approach to comparing the different model structures. There is well-developed theory of how to include effects of penalization and smoothing parameter uncertainty when estimating the model complexity penalty for AIC (*Wood, Pya & Säfken, 2016*). We demonstrate this approach in Table 1. Using AIC, there is strong support for including among-group functional variability for both the `CO2` dataset and the `bird_move` dataset (compare models $G$ vs. all other models). For the `CO2` dataset (Table 1A), there is relatively strong evidence that there is more intergroup variability in smoothness than model $GS$ allows, and weaker evidence that model $S$ or $I$ (separate smoothers for all plants) show the best fit.

For the `bird_move` dataset (Table 1B), model $GS$ (global smoother plus group-level smoothers with a shared penalty) gives the best fit for all models including a global smooth (which is good as we simulated the data from a model with this structure!). However, model $S$ (without a global term) still fits this data better than model $GS$ based on AIC. This highlights an issue with AIC for selecting between models with and without a global smooth: as it is possible to fully recreate the global term by just allowing each group-level smoother to have a similar shape to one another (i.e., the global term is totally concurve with the group-level smoothers; see section "Computational and Statistical Issues When Fitting HGAMs") model selection criteria such as AIC may indicate that the extra parameters required to fit the global smoother are unnecessary[5].

Given this issue with selecting global terms, we strongly recommend not selecting models based purely on AIC. Instead, model selection should be based on expert subject knowledge about the system, computational time, and most importantly, the inferential goals of the study. Table 1A indicates that models $S$ and $I$ (which do not have a global function) fit the `CO2` data better than models with a global function, and that

[5] If it is important for a given study to determine if there is evidence for a significant global smooth effect, we recommend fitting model $GS$ or $GI$, including the argument `select = TRUE` in the `gam` function. This has the effect of adding an extra penalty to each smooth term, that penalizes functions in the null space of the penalty matrices for each smooth. By doing this, it is possible for `mgcv` to penalize all model terms to a zero effect, in effect doing variable selection (*Marra & Wood, 2011*). When `select=TRUE`, the significance of the global term can be found by looking at the significance of the term in `summary.gam(model)`. Note that this can significantly increase the amount of time it takes to fit a model for data sets with a large number of penalty terms (such as model $GI$ when the number of groups is high).

model *S* fits the `bird_move` data better than model *GS*. However, it is the shape of the global function that we are actually interested in here, as models *S* and *I* cannot be used to predict the concentration-uptake relationship for plants that are not part of the training set, or the average migration path for birds. The same consideration holds when choosing between model *GS* and *GI*: while model *GI* fits the `CO2` data better than model *GS* (as measured by AIC), model *GS* can be used to simulate functional variation for unobserved group levels, whereas this is not possible within the framework of model *GI*. The next section works through two examples to show how to choose between different models, and section "Computational and Statistical Issues When Fitting HGAMs" discusses these and other model fitting issues in more depth.

It also is important to recognize that AIC, like any function of the data, is a random variable and should be expected to have some sampling error (*Forster & Sober, 2011*). In cases when the goal is to select the model that has the best predictive ability, we recommend holding some fraction of the data out prior to the analysis and comparing how well different models fit that data, or using *k*-fold cross validation as a more accurate guide to how well a given model may predict out of sample. Predictive accuracy may also be substantially improved by averaging over multiple models (*Dormann et al., 2018*).

## EXAMPLES

We now demonstrate two worked examples on one data set to highlight how to use HGAMs in practice, and to illustrate how to fit, test, and visualize each model. We will demonstrate how to use these models to fit community data, to show when using a global trend may or may not be justified, and to illustrate how to use these models to fit seasonal time series.

For these examples, data are from a long-term study in seasonal dynamics of zooplankton, collected by the Richard Lathrop. The data were collected from a chain of lakes in Wisconsin (Mendota, Monona, Kegnonsa, and Waubesa) approximately bi-weekly from 1976 to 1994. They consist of samples of the zooplankton communities, taken from the deepest point of each lake via vertical tow. The data are provided by the Wisconsin Department of Natural Resources and their collection and processing are fully described in *Lathrop (2000)*.

Zooplankton in temperate lakes often undergo seasonal cycles, where the abundance of each species fluctuates up and down across the course of the year, with each species typically showing a distinct pattern of seasonal cycles. The inferential aims of these examples are to (i) estimate variability in seasonality among species in the community in a single lake (Mendota), and (ii) estimate among-lake variability for the most abundant taxon in the sample (*Daphnia mendotae*) across the four lakes. To enable evaluation of out-of-sample performance, we split the data into testing and training sets. As there are multiple years of data, we used data from the even years to fit (train) models, and the odd years to test the fit.

Each record consists of counts of a given zooplankton taxon taken from a subsample from a single vertical net tow, which was then scaled to account for the relative volume of subsample vs. the whole net sample and the area of the net tow, giving population
[6] A more appropriate model for this data would be to assume that density is *left censored*, where 1,000 is treated as a threshold which the data may lie below, but it is not possible to measure lower than this. However, **mgcv** does not currently have a left-censored family. The **brms** package, for Bayesian model fitting, can fit a left-censored Gamma distribution, so it would be possible to fit this model using that software. We discuss using HGAMs in **brms** in the section "Computational and Statistical Issues When Fitting HGAMs."

density per $m^2$. Values are rounded to the nearest 1,000. Observed densities span four orders of magnitude. We modelled density using a Gamma distribution with a log-link. For any net tow sample where a given taxon was not observed, we set that taxon's density to 1,000 (the minimum possible sample size)[6]. To evaluate how well each model fits new data (not used to fit the model), we calculated the total deviance of the out-of-sample data that we had previously held out. The deviance is equal to two times the sum of the difference between the log-likelihood of the out-of-sample data (as predicted by each model) and a saturated model that has one predictor for each data point, all multiplied by the scale parameter for the family of interest. It can be interpreted similarly to the residual sum of squares for a simple linear regression (*Wood, 2017a*, p. 109).

First, we demonstrate how to model community-level variability in seasonality, by regressing scaled density on day of year with species-specific curves. As we are not interested in average seasonal dynamics, we focus on models $S$ and $I$ (if we wanted to estimate the seasonal dynamics for rarer species, adding a global smooth term might be useful, so we could borrow information from the more common species). As the data are seasonal, we use cyclic smoothers as the basis for seasonal dynamics. Therefore, we need to specify start and end points for our cycles using the `knots` argument to `gam()`, as well as specify this smoother type as a factor-smooth interaction term using the `xt` argument (the `xt` argument is how any extra information that a smoother might need is supplied; see `?mgcv::s` for more information). Note that we also include a random effect smoother for both `taxon` and `taxon:year_f`, where `year_f` is `year` transformed into a factor variable. This deals with the fact that average zooplankton densities can show large year-to-year variation. The argument `drop.unused.levels=FALSE` is also included so the `gam` function does not drop the year factor levels corresponding to those in the held-out test data set.

### Model S

```
zoo_comm_modS <- gam(density_adj ~ s(taxon, year_f, bs="re") +
                s(day, taxon, bs="fs", k=10, xt=list(bs="cc")),
                data=zoo_train, knots=list(day=c(0, 365)),
                family=Gamma(link="log"), method="REML",
                drop.unused.levels=FALSE)
```

### Model I

```
# Note that s(taxon, bs="re") has to be explicitly included here, as the
# day by taxon smoother does not include an intercept
zoo_comm_modI <- gam(density_adj ~ s(day, by=taxon, k=10, bs="cc") +
                s(taxon, bs="re") + s(taxon, year_f, bs="re"),
                data=zoo_train, knots=list(day=c(0, 365)),
                family=Gamma(link="log"), method="REML",
                drop.unused.levels=FALSE)
```

At this stage of the analysis (prior to model comparisons), it is useful to determine if any of the fitted models adequately describe patterns in the data (i.e., goodness of fit testing).

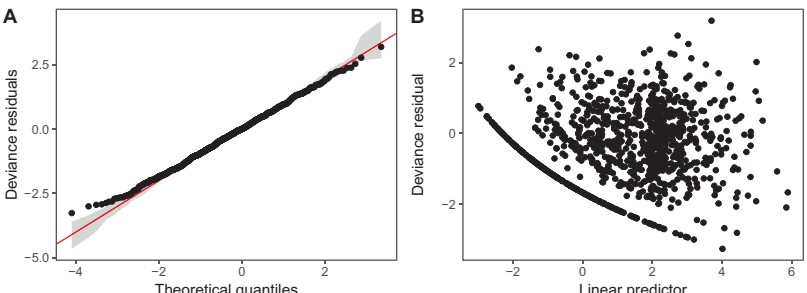

**Figure 14 Diagnostic plots for model *I* fitted to zooplankton community data in Lake Mendota.** (A) QQ-plot of residuals (black). Red line indicates the 1–1 line and gray bands correspond to the expected 95% CI for the QQ plot, assuming the distribution is correct. (B) Deviance residuals vs. fitted values (on the link scale).

mgcv's `gam.check()` facilitates this model-checking. This function creates a set of standard diagnostic plots: a QQ plot of the deviance residuals (see *Wood (2017a)*) compared to their theoretical expectation for the chosen family, a plot of response vs. fitted values, a histogram of residuals, and a plot of residuals vs. fitted values. It also conducts a test for each smooth term to determine if the number of degrees of freedom (`k`) for each smooth is adequate (see `?mgcv::gam.check` for details on how that test works). The code for checking model *S* and *I* for the community zooplankton model is:

```
gam.check(zoo_comm_modS)
gam.check(zoo_comm_modI)
```

We have plotted QQ plots and fitted-vs. residual plots for model *I* (fitted vs. response plots are generally less useful for non-normally distributed data as it can be difficult to visually assess if the observed data shows more heteroskedasticity than expected). The results for model *S* are virtually indistinguishable to the naked eye. We have also used alternate QQ-plotting code from the **gratia** package (*Simpson, 2018*), using the `qq_plot()` function, as this function creates a **ggplot2** object that are easier to customize than the **base** plots from `gam.check()`. The code for generating these plots is in the Supplemental Material. These plots (Fig. 14) indicate that the Gamma distribution seems to fit the observed data well except at low values, where the deviance residuals are larger than predicted by the theoretical quantiles (Fig. 14A). There also does not seem to be a pattern in the residual vs. fitted values (Fig. 14B), except for a line of residuals at the lowest values, which correspond to all of those observations where a given taxon was absent from the sample.

The `k.check()` test (Table 2) shows that the default maximum degrees of freedom for the smoothers used in model *I* are sufficient for all species, as the EDF for all estimated smoothers are well below their maximum possible value ($k'$), and the $p$-value for the observed $k$-index (which measures pattern in the residuals) is not significant.

In this table, each row corresponds to a single smooth term, $k'$ corresponds to the number of basis functions used for that smoother in the fitted model (smaller than the specified `k` in the model itself, as some basis functions are automatically dropped to ensure the model is identifiable). The column EDF is the estimated EDF for that smoother,

**Table 2 Results from running** `k.check()` **on** `zoo_comm_modI`.

| Model term | $k'$ | EDF | $k$-index | $p$-value |
|---|---|---|---|---|
| s(day):taxon*C. sphaericus* | 8 | 4.78 | 0.89 | 0.44 |
| s(day):taxon*Calanoid copepods* | 8 | 6.66 | 0.89 | 0.46 |
| s(day):taxon*Cyclopoid copepods* | 8 | 5.31 | 0.89 | 0.46 |
| s(day):taxon*D. mendotae* | 8 | 6.95 | 0.89 | 0.46 |
| s(day):taxon*D. thomasi* | 8 | 6.57 | 0.89 | 0.45 |
| s(day):taxon*K. cochlearis* | 8 | 5.92 | 0.89 | 0.47 |
| s(day):taxon*L. siciloides* | 8 | 0.52 | 0.89 | 0.46 |
| s(day):taxon*M. edax* | 8 | 4.69 | 0.89 | 0.43 |
| s(taxon) | 8 | 6.26 | NA | NA |
| s(taxon,year_f) | 152 | 51.73 | NA | NA |

**Note:**

Each row corresponds to a single model term. The notation for term names uses `mgcv` syntax. For instance, "s(day):taxon*C. sphaericus*" refers to the smoother for day for the taxon *C. sphaericus*.

the $k$-index is a measure of the remaining pattern in the residuals, and the $p$-value is calculated based on the distribution of the $k$-index after randomizing the order of the residuals. Note that there is no $p$-value for the random effects smoothers `s(taxon)` and `s(taxon,year_f)` as the $p$-value is calculated from simulation-based tests for autocorrelation of the residuals. As `taxon` and `year_f` are treated as simple random effects with no natural ordering, there is no meaningful way of checking for autocorrelation.

Differences between models *S* (shared smoothness between taxa) and *I* (different smoothness for each taxa) seem to be driven by the low seasonality of *Leptodiaptomus siciloides* relative to the other species, and how this is captured by the more flexible model *I* (Fig. 15). Still, both models show very similar fits to the training data. This implies that the added complexity of different penalties for each species (model *I*) is unnecessary here, which is consistent with the fact that model *S* has a lower AIC (4667) than model *I* (4677), and that model *S* is somewhat better at predicting out-of-sample fits for all taxa than model *I* (Table 3). Both models show significant predictive improvement compared to the intercept-only model for all species except *Keratella cochlearis* (Table 3). This may be driven by changing timing of the spring bloom for this species between training and out-of-sample years (Fig. 15).

Next, we look at how to fit interlake variability in dynamics for just *Daphnia mendotae*. Here, we will compare models *G*, *GS*, and *GI* to determine if a single global function is appropriate for all four lakes, or if we can more effectively model variation between lakes with a shared smoother and lake-specific smoothers.

## Model G

```
zoo_daph_modG <- gam(density_adj ~ s(day, bs="cc", k=10) + s(lake, bs="re") +
                        s(lake, year_f, bs="re"),
                    data=daphnia_train, knots=list(day=c(0, 365)),
                    family=Gamma(link="log"), method="REML",
                    drop.unused.levels=FALSE)
```

## Model GS

```
zoo_daph_modGS <- gam(density_adj ~ s(day, bs="cc", k=10) +
                        s(day, lake, k=10, bs="fs", xt=list(bs="cc")) +
                        s(lake, year_f, bs="re"),
                    data=daphnia_train, knots=list(day=c(0, 365)),
                        family=Gamma(link="log"), method="REML",
                        drop.unused.levels=FALSE)
```

## Model GI

```
zoo_daph_modGI <- gam(density_adj~s(day, bs="cc", k=10) +s(lake, bs="re") +
                        s(day, by=lake, k=10, bs="cc") +
                        s(lake, year_f, bs="re"),
                    data=daphnia_train, knots=list(day=c(0, 365)),
                        family=Gamma(link ="log"), method="REML",
                        drop.unused.levels=FALSE)
```

Diagnostic plots from `gam.check()` indicate that there are no substantial patterns comparing residuals to fitted values (not shown), and QQ-plots are similar to those from the zooplankton community models; the residuals for all three models closely correspond to the expected (Gamma) distribution, except at small values, where the observed residuals are generally larger than expected (Fig. 16). As with the community data, this is likely an artifact of the assumption we made of assigning zero observations a value of 1,000 (the lowest possible value), imposing an artificial lower bound on the observed counts. There was also some evidence that the largest observed values were smaller than expected given the theoretical distribution, but these fell within the 95% CI for expected deviations from the 1–1 line (Fig. 16).

AIC values indicate that both model *GS* (1,093.71) and *GI* (1,085.7) are better fits than model *G* (1,097.62), with model *GI* fitting somewhat better than model *GS*.[7] There does not seem to be a large amount of interlake variability (the EDF per lake are low in models *GS* & *GI*). Plots for all three models (Fig. 17) show that Mendota, Monona, and Kegonsa lakes are very close to the average and to one another for both models, but Waubesa shows evidence of a more pronounced spring bloom and lower winter abundances.

Model *GI* is able to predict as well or better than model *G* or *GS* for all lakes (Table 4), indicating that allowing for interlake variation in seasonal dynamics improved model prediction. All three models predicted dynamics in Lake Mendota and Lake Menona significantly better than the intercept-only model (Table 4). None of the models did well in terms of predicting Lake Waubesa dynamics out-of-sample compared to a simple model with only a lake-specific intercept and no intra-annual variability, but this was due to the influence of a single large outlier in the out-of-sample data that occurred after the spring bloom, at day 243 (Fig. 17; note that the *y*-axis is log-scaled). However, baring a more detailed investigation into the cause of this large value, we cannot arbitrarily exclude this outlier from the goodness-of-fit analysis; it may be due either to measurement error or a true high late-season *Daphnia* density that our model was not able to predict.

[7] When comparing models via AIC, we use the standard rule of thumb from *Burnham & Anderson (1998)*, where models that differ by two units or less from the lowest AIC model have substantial support, and those differing by more than four units have less support.

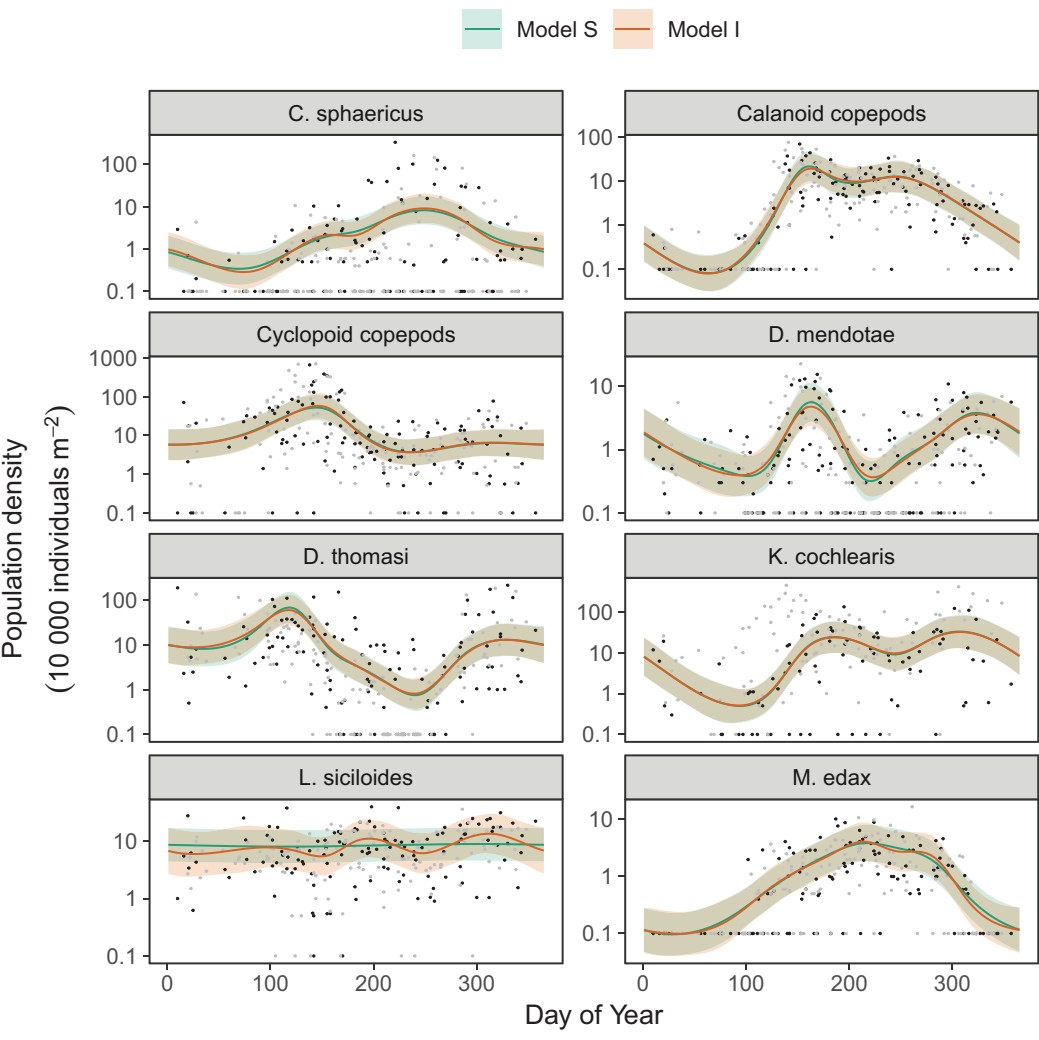

**Figure 15 Species-specific seasonal dynamics for the eight zooplankon species tracked in Lake Mendota.** Black points indicate individual plankton observations in the training data, and gray points are observations in held-out years used for model validation. Lines indicate predicted average values for model *S* (green) and model *I* (red). Ribbons indicate ±2 standard errors around the mean.

# COMPUTATIONAL AND STATISTICAL ISSUES WHEN FITTING HGAMS

Which of the five model formulations (Fig. 4) should you choose for a given data set? There are two major trade-offs to consider. The first is the bias-variance trade-off: more complex models can account for more fluctuations in the data, but also tend to give more variable predictions, and can overfit. The second trade-off is model complexity vs. computational cost: more complex models can include more potential sources of variation and give more information about a given data set, but will generally take more time and computational resources to fit and debug. We discuss both of these trade-offs in this section. We also discuss how to extend the HGAM framework to fit more complex models.

**Table 3 Out-of-sample predictive ability for model *S* and *I* applied to the zooplankton community dataset.**

| Taxon | Total deviance of out-of-sample data | | |
|---|---|---|---|
| | Intercept only | Model *S* | Model *I* |
| *C. sphaericus* | 715 | 482 | 495 |
| *Calanoid copepods* | 346 | 220 | 223 |
| *Cyclopoid copepods* | 569 | 381 | 386 |
| *D. mendotae* | 353 | 264 | 268 |
| *D. thomasi* | 486 | 333 | 337 |
| *K. cochlearis* | 486 | 2260 | 2340 |
| *L. siciloides* | 132 | 116 | 126 |
| *M. edax* | 270 | 138 | 139 |

Note:
Deviance values represent the total deviance of model predictions from observations for out-of-sample data. "Intercept only" results are for a null model with only taxon-level random effect intercepts included.

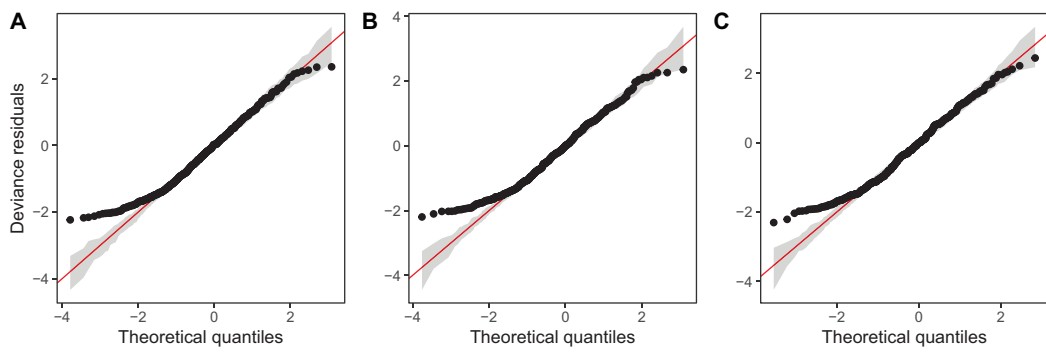

**Figure 16 QQ-plots for model *G* (A), *GS* (B), and *GI* (C) fitted to Daphnia data across the four lakes.** Red line indicates the 1-1 line, black points are observed model residuals, and gray bands correspond to the expected 95% CI for the QQ plot, assuming the distribution is correct.

## Bias-variance trade-offs

The bias-variance trade-off is a fundamental concept in statistics. When trying to estimate any relationship (in the case of GAMs, a smooth relationship between predictors and data) bias measures how far, on average, an estimate is from the true value. The variance of an estimator corresponds to how much that estimator would fluctuate if applied to multiple different samples of the same size taken from the same population. These two properties tend to be traded off when fitting models. For instance, rather than estimating a population mean from data, we could simply use a predetermined fixed value regardless of the observed data[8]. This estimate would have no variance (as it is always the same regardless of what the data look like) but would have high bias unless the true population mean happened to equal the fixed value we chose. Penalization is useful because using a penalty term slightly increases model bias, but can substantially decrease variance (*Efron & Morris, 1977*).

In GAMs, the bias-variance trade-off is managed by the terms of the penalty matrix, and equivalently random effect variances in HGLMs. Larger penalties correspond to lower

[8] While this example may seem contrived, this is exactly what happens when we assume a given regression coefficient is equal to zero (and thus exclude it from a model).

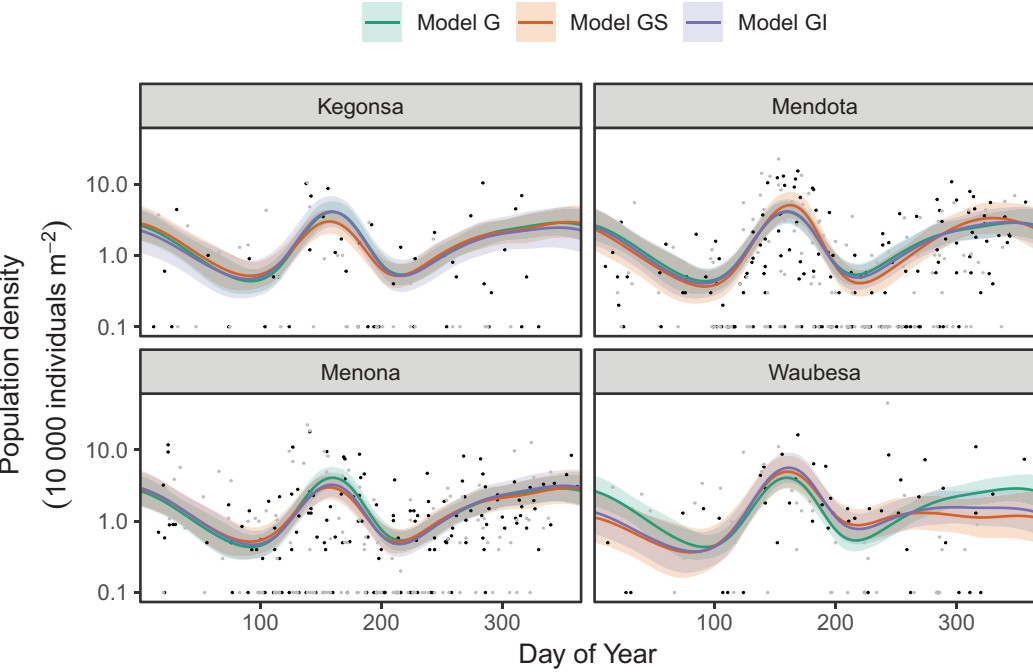

**Figure 17 Raw data (points) and fitted models (lines) for *D. mendota* data.** Black points indicate individual plankton observations in the training data, and gray points are observations in held-out years used for model validation. Green line: model *G* (no inter-lake variation in dynamics); orange line: model *GS* (interlake variation with similar smoothness); purple line: model *GI* (varying smoothness among lakes). Shaded bands are drawn at ±2 standard errors around each model.

**Table 4 Out-of-sample predictive ability for model *G*, *GS*, and *GI* applied to the *D. mendotae* dataset.**

| Lake | Total deviance of out-of-sample data | | | |
|---|---|---|---|---|
| | Intercept only | Model *G* | Model *GS* | Model *GI* |
| Kegonsa | 96 | 92 | 89 | 86 |
| Mendota | 352 | 258 | 257 | 257 |
| Menona | 348 | 300 | 294 | 290 |
| Waubesa | 113 | 176 | 164 | 157 |

Note:
Deviance values represent the total deviance of model predictions from observations for out-of-sample data. "Intercept only" results are for a null model with only lake-level random effect intercepts included.

variance, as the estimated function is unable to wiggle a great deal, but also correspond to higher bias unless the true function is close to the null space for a given smoother (e.g., a straight line for TPRS with second derivative penalties, or zero for a random effect). The computational machinery used by **mgcv** to fit smooth terms is designed to find penalty terms that best trade-off bias for variance to find a smoother that can effectively predict new data.

The bias-variance trade-off comes into play with HGAMs when choosing whether to fit separate penalties for each group level or assign a common penalty for all group levels (i.e., deciding between models *GS* & *GI* or models *S* & *I*). If the functional relationships we are

trying to estimate for different group levels actually vary in how wiggly they are, setting the penalty for all group-level smoothers equal (models *GS* & *S*) will either lead to overly variable estimates for the least variable group levels, over-smoothed (biased) estimates for the most wiggly terms, or a mixture of these two, depending on the fitting criteria.

We developed a simple numerical experiment to determine whether **mgcv**'s fitting criteria tend to set estimated smoothness penalties high or low in the presence of among-group variability in smoothness when fitting model *GS* or *S* HGAMs. We simulated data from four different groups, with all groups having the same levels of the covariate $x$, equally spaced across the range from 0 to $2\pi$. For each group, the true function relating $x$ to the response, $y$, was a cosine wave, but the frequency varied from 0.5 (equal to half a cycle across the range of $x$) to 4 (corresponding to 4 full cycles across the range). As all four sine waves spanned the whole range from $-1$ to $+1$ across the range of $x$, and as they were all integer or half-integer frequencies, the signal for all groups had the same variance across the range of $x$, approximately equal to 0.5. Therefore, the true function for all groups had the same strength of signal; all that varied between groups was how rapidly the signal fluctuated. We added normally distributed error to all $y$-values, with three different noise levels, given by standard deviations of 0.5, 1, and 2. These correspond to signal-to-noise ratios (i.e., variance of the cosine curve divided by variance of the noise) of 2, 0.5, and 0.125. For each noise level we created 25 replicate data sets to illustrate the amount of simulation-to-simulation variation in model fit. We then fit both model *S* (where all curves were assumed to be equally smooth) and model *I* (with varying smoothness) to each replicate for each noise level, using REML criteria to estimate penalties.

A sample of the fits for each group for three of the replicates for each model are shown in Fig. 18A, with model *S* in red and model *I* in blue. Figure 18B illustrates how well each model fared across the range of replicates at accurately estimating the true smoothness of the highest frequency terms as measured by the squared second derivative of the smooth fit vs. that of the true function, with the distance to the black one-to-one line indicating the degree to which the estimated function for each group over- or under-estimated the smoothness of the true signal. In general, under low noise conditions (Fig. 18, signal-to-noise ratio of 2), model *S* tended to overfit the smoothest, lowest-frequency, groups, while accurately fitting the highest frequency groups. Under moderate signal-to-noise ratios, model *S* tended to over-penalize high-frequency groups and under-penalize low frequency groups, and in the lowest signal-to-noise ratio tested (0.125), model *S* tended to penalize all groups toward very smooth functions (Fig. 18B). Curves estimated by model *I*, on the other hand, tended to accurately capture the true wiggliness of the function across the whole range of frequencies and noises, except for the lowest-frequency groups, and the highest frequency groups it the presence of high noise; in both cases, model *I* tended to over-smooth (Fig. 18B).

This implies that assuming equal smoothness will result in underestimating the true smoothness of low-variability terms in cases of high signal-to-noise, and overestimating the true smoothness of high-frequency terms in low signal-to-noise data sets. If this is a potential issue, we recommend fitting both models *S* and *I* and using standard model evaluation criteria (e.g., AIC) or out-of-sample predictive accuracy (as in the section

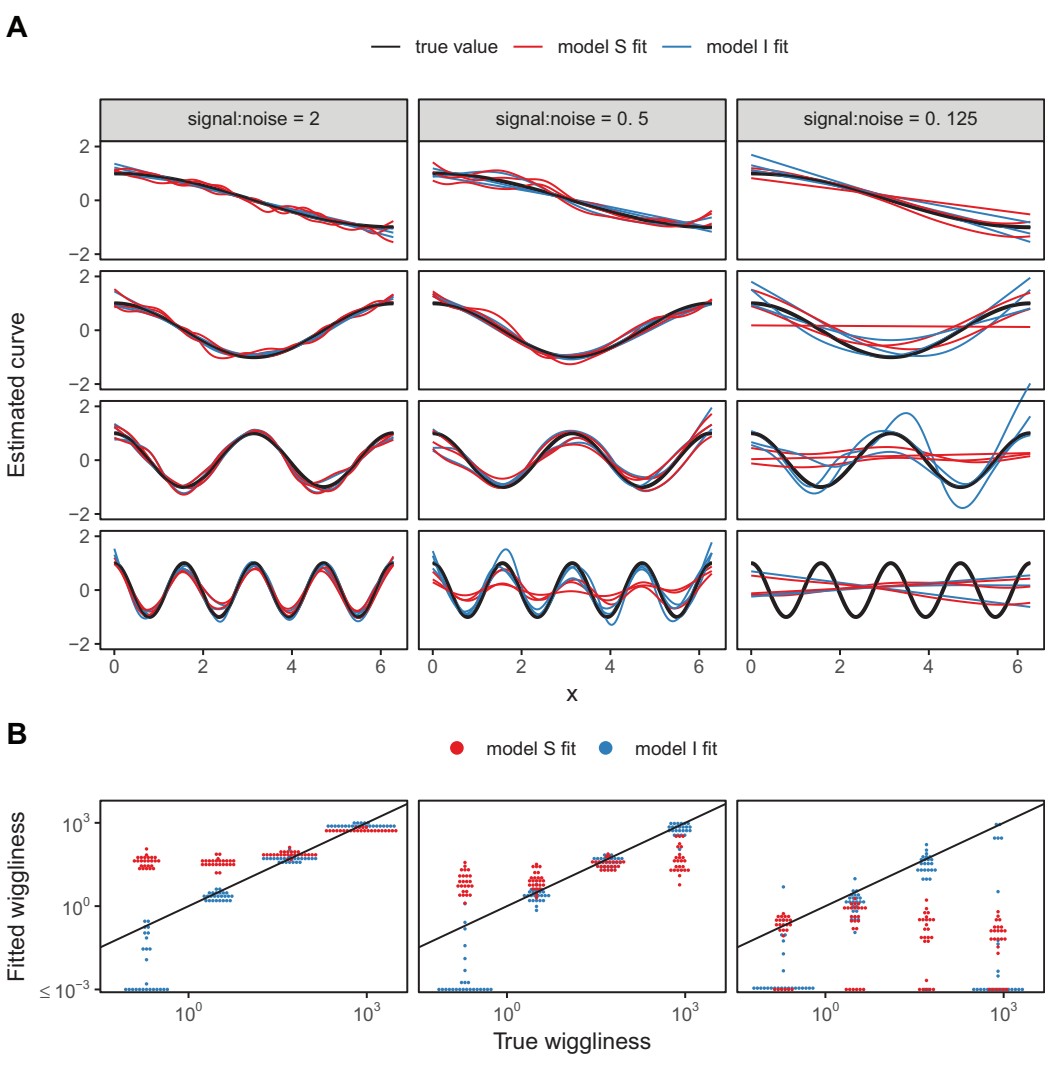

**Figure 18** (A) Illustration of bias that can arise from assuming equal smoothness for all group levels (model *S*, red lines) vs. allowing for intergroup variation in smoothness (model *I*, blue lines) across a range of signal-to-noise ratios, holding the group-level signals constant. The true function for each group level is shown in black. (B) Distribution of wiggliness (as measured by the integral of the squared second derivative) of the estimated function for each replicate for each group level for model *S* (red) and model *I* (blue), vs. the true wiggliness of the function for that grouping level, with the black line indicating the one-to-one line. Points below (above) the black line indicate that a given model estimated the curve as less (more) wiggly than the true curve used to generate the data. Estimated wiggliness less than $10^{-3}$ was truncated for visual clarity, as **mgcv** estimated effectively straight lines for several groups, corresponding to a wiggliness of 0, which would not appear on a log-scaled plot.

"Examples") to determine if there is evidence for among-group variability in smoothness. However, it may be the case that there are too few data points per group to estimate separate smoothness levels, in which case model *GS* or model *S* may still be the better option even in the face of varying smoothness.

The ideal case would be to assume that among-group penalties follow their own distribution (estimated from the data), to allow variation in smoothness while still getting the benefit of pooling information on smoothness between groups. This is currently not

implemented in **mgcv**. It is possible to set up this type of varying penalty model in flexible Bayesian modeling software such as **Stan** (see below for a discussion of how to fit HGAMs using these tools), where intergroup variation in smoothing penalties could be modelled with a hierarchical prior. However, to the best of our knowledge, how to fit this type of model has not been well studied in either the Bayesian or frequentist literature.

It may seem there is also a bias-variance trade-off between choosing to use a single global smoother (model *G*) or a global smoother plus group-level terms (models *GS* and *GI*). In model *G*, all the data is used to estimate a single smooth term, and thus should have lower variance than models *GS* and *GI*, but higher bias for any given group in the presence of intergroup functional variability. However, in practice, this trade-off will be handled via penalization; if there are no average differences between functional responses, **mgcv** will penalize the group-specific functions toward zero, and thus toward the global model. The choice between using model *G* vs. models *GS* and *GI* should generally be driven by computational costs. Model *G* is typically much faster to fit than models *GS* and *GI*, even in the absence of among-group differences. If there is no need to estimate intergroup variability, model *G* will typically be more efficient.

A similar issue exists when choosing between models *GS* and *GI* and models *S* and *I*. If all group levels have very different functional shapes, the global term will get penalized toward zero in models *GS* and *GI*, so they will reduce to models *S* and *I*. The choice to include a global term should be made based on scientific considerations (is the global term of interest?) and computational considerations.

## Complexity-computation trade-offs

The more flexible a model is, the larger an effective parameter space any fitting software has to search. It can be surprisingly easy to use massive computational resources trying to fit models to even small datasets. While we typically want to select models based on their fit and our inferential goals, computing resources can often act as an effective upper bound on model complexity. For a given data set, assuming a fixed family and link function, the time taken to estimate an HGAM will depend (roughly) on four factors: (i) the number of coefficients to be estimated (and thus the number of basis functions chosen), (ii) the number of smoothing parameters to be estimated, (iii) whether the model needs to estimate both a global smoother and group-level smoothers, and (iv) the algorithm and fitting criteria used to estimate parameters.

The most straightforward factor that will affect the amount of computational resources is the number of parameters in the model. Adding group-level smoothers (moving from model *G* to the other models) means that there will be more regression parameters to estimate. For a dataset with $g$ different groups and $n$ data points, fitting a model with just a global smoother, `y~s(x,k=k)` will require $k$ coefficients, and takes $\mathcal{O}(nk^2)$ operations to evaluate. Fitting the same data using a group-level smoother (model *S*, `y~s(x,fac,bs="fs",k=k)`) will require $\mathcal{O}(nk^2g^2)$ operations to evaluate. In effect, adding a group-level smoother will increase computational cost by an order of the number of groups squared. The effect of this is visible in the examples we fit in the section "What are Hierarchical GAMs?." Table 5 compares the relative time it takes to compute model *G* vs. the other models.

**Table 5 Relative computational time and model complexity for different HGAM formulations of the two example data sets from section "What are hierarchical GAMs?".**

| Model | Relative time | Number of terms | |
|---|---|---|---|
| | | Coefficients | Penalties |
| A. CO2 data | | | |
| G | 1 | 17 | 2 |
| GS | 7 | 65 | 3 |
| GI | 14 | 65 | 14 |
| S | 5 | 61 | 3 |
| I | 16 | 61 | 13 |
| B. Bird movement data | | | |
| G | 1 | 90 | 2 |
| GS | 510 | 540 | 8 |
| GI | 390 | 624 | 14 |
| S | 820 | 541 | 6 |
| I | 70 | 535 | 12 |

Note:
All times are scaled relative to the length of time model *G* takes to fit to that data set. The number of coefficients measures the total number of model parameters (including intercepts). The number of smoothers is the total number of unique penalty values estimated by the model.

One way to deal with this issue would be to reduce the number of basis functions used when fitting group-level smoothers when the number of groups is large, limiting the flexibility of the model. It can also make sense to use more computationally-efficient basis functions when fitting large data sets, such as P-splines (*Wood, 2017b*) or cubic splines. TPRSs entail greater computational costs (*Wood, 2017a*).

Including a global smoother (models *GS* and *GI* compared to models *S* and *I*) will not generally substantially affect the number of coefficients that need to be estimated (Table 5). Adding a global term will add at most k extra terms. It can be substantially less than that, as **mgcv** drops basis functions from co-linear smoothers to ensure that the model matrix is full rank.

Adding additional smoothing parameters (moving from model *GS* to *GI*, or moving from model *S* to *I*) is more costly than increasing the number of coefficients to estimate, as estimating smoothing parameters is computationally intensive (*Wood, 2011*). This means that models *GS* and *S* will generally be substantially faster than *GI* and *I* when the number of groups is large, as models *GI* and *I* fit a separate set of penalties for each group level. The effect of this is visible in comparing the time it takes to fit model *GS* to model *GI* (which has a smoother for each group) or models *S* and *I* for the CO2 example data (Table 5). Note that this will not hold in all cases. For instance, model *GI* and *I* take less time to fit the bird movement data than models *GS* or *S* do (Table 5B).

## Alternative formulations: `bam()`, `gamm()`, and `gamm4()`

When fitting models with large numbers of groups, it is often possible to speed up computation substantially by using one of the alternative fitting routines available through **mgcv**.

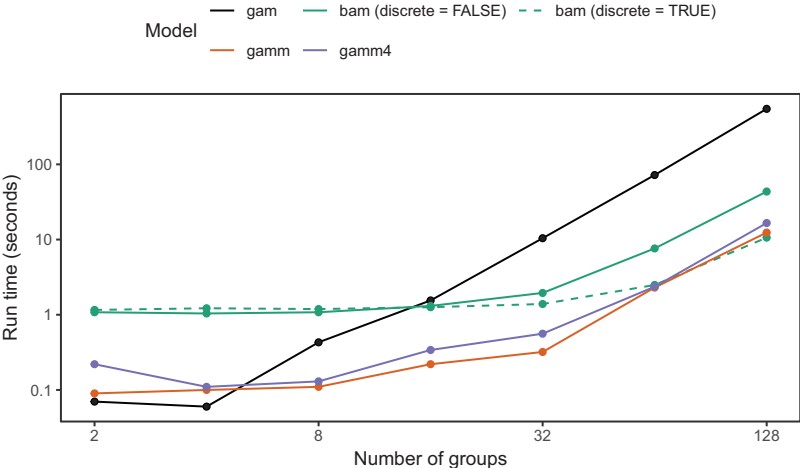

**Figure 19 Elapsed time to estimate the same model using each of the four approaches.** Each data set was generated with 20 observations per group using a unimodal global function and random group-specific functions consisting of an intercept, a quadratic term, and logistic trend for each group. Observation error was normally distributed. Models were fit using model 2: `y s(x, k=10, bs="cp") + s(x,fac, k=10, bs="fs", xt=list(bs="cp"), m=1)`. All models were run on a single core.

The first option is the function `bam()`, which requires the fewest changes to existing code written using the `gam()` function. `bam()` is designed to improve performance when fitting large data sets via two mechanisms. First, it saves on memory needed to compute a given model by using a random subset of the data to calculate the basis functions. It then blocks the data and updates model fit within each block (*Wood, Goude & Shaw, 2015*). While this is primarily designed to reduce memory usage, it can also substantially reduce computation time. Second, when using `bam()`'s default `method="fREML"` ("Fast REML") method, you can use the `discrete=TRUE` option: this first bins continuous covariates into a smaller number of discrete values before estimating the model, substantially reducing the amount of computation needed (*Wood et al., 2017*; see `?mgcv::bam` for more details). Setting up the five model types (Fig. 4) in `bam()` uses the same code as we have previously covered; the only difference is that you use the `bam()` instead of `gam()` function, and have the additional option of discretizing your covariates.

`bam()` has a larger computational overhead than `gam()`; for small numbers of groups it can be slower than `gam()` (Fig. 19). As the number of groups increases, computational time for `bam()` increases more slowly than for `gam()`; in our simulation tests, when the number of groups is greater than 16, `bam()` can be upward of an order of magnitude faster (Fig. 19). Note that `bam()` can be somewhat less computationally stable when estimating these models (i.e., less likely to converge). While base `bam()` (not fit using `discrete=TRUE`) is slower than the other approaches shown in Fig. 19, that does not imply that `bam()` is a worse choice in general; it is designed to avoid memory limitations when working with big data rather than explicitly speeding up model fitting. The `bam()` functions would likely show much better relative performance when the number of

individuals per group were large (in the hundreds to thousands, compared to the 20 individuals per group used in Fig. 19).

The second option is to fit models using one of two dedicated mixed effect model estimation packages, **nlme** and **lme4**. The **mgcv** package includes the function gamm(), which uses the **nlme** package to estimate the GAM, automatically handling the transformation of smooth terms into random effects (and back into basis function representations for plotting and other statistical analyses). The gamm4() function, in the separate **gamm4** package, uses **lme4** in a similar way. Using gamm() or gamm4() to fit models rather than gam() can substantially speed up computation when the number of groups is large, as both **nlme** and **lme4** take advantage of the sparse structure of the random effects, where most basis functions will be zero for most groups (i.e., any group-specific basis function will only take a nonzero value for observations in that group level). As with bam(), gamm(), and gamm4() are generally slower than gam() for fitting HGAMs when the number of group levels is small (in our simulations, <8 group levels), however they do show substantial speed improvements even with a moderate number of groups, and were as fast as or faster to calculate than bam() for all numbers of grouping levels we tested (Fig. 19)[9].

Both gamm() and gamm4() require a few changes to model code. First, there are a few limitations on how you are able to specify the different model types (Fig. 4) in both frameworks. Factor-smoother interaction (bs="fs") basis setup works in both gamm() and gamm4(). However, as the **nlme** package does not support crossed random effects, it is not possible to have two factor-smoother interaction terms for the same grouping variable in gamm() models (e.g., y~s(x1, grp, bs="fs")+s(x2, grp, bs="fs"). These type of crossed random effects are allowed in **gamm4**. The use of te() terms are not possible in **gamm4**, due to issues with how random effects are specified in the **lme4** package, making it impossible to code models where multiple penalties apply to a single basis function. Instead, for multidimensional group-level smoothers, the alternate function t2() needs to be used to generate these terms, as it creates tensor products with only a single penalty for each basis function (see ?mgcv::t2 for details on these smoothers, and *Wood, Scheipl & Faraway (2013)* for the theoretical basis behind this type of tensor product). For instance, model *GS* for the bird movement data we discussed in the section "What are Hierarchical GAMs?" would need to be coded as:

```
bird_modS <- gamm4(count ~ t2(week, latitude, species, k=c(10, 10, 6), m=2,
                       bs=c("cc", "tp", "re")),
                   data=bird_move, family="poisson")
```

These packages also do not support the same range of families for the dependent variable; gamm() only supports non-Gaussian families by using a fitting method called penalized quasi-likelihood, that is, slower and not as numerically stable as the methods used in gam(), bam(), and gamm4(). Non-Gaussian families are well supported by **lme4** (and thus **gamm4**), but can only fit them using ML rather than REML, so may tend to over-smooth relative to gam() using REML estimation. Further, neither gamm() nor gamm4() supports several of the extended families available through **mgcv**,

[9] It is also possible to speed up both gam() and bam() by using multiple processors in parallel, whereas this is not currently possible for gamm() and gamm4(). For large numbers of grouping levels, this should speed up computation as well, at the cost of using more memory. However, computation time will likely not decline linearly with the number of cores used, since not all model fitting steps are parallelizable, and performance of cores can vary. As parallel processing can be complicated and dependent on the type of computer you are using to configure, we do not go into how to use these methods here. The help file ?mgcv::mgcv.parallel explains how to use parallel computations for gam() and bam() in detail.

such as zero-inflated, negative binomial, or ordered categorical and multinomial distributions.

## Estimation issues when fitting both global and group-level smoothers

When fitting models with separate global and group-level smoothers (models *GS* and *GI*), one issue to be aware of is concurvity between the global smoother and group-level terms. Concurvity measures how well one smooth term can be approximated by some combination of the other smooth terms in the model (see `?mgcv::concurvity` for details). For models *GS* and *GI*, the global term is either entirely or almost entirely[10] concurved with the group-level smoothers. This is because, in the absence of the global smooth term, it would be possible to recreate that average effect by shifting all the group-level smoothers so they were centered around the global mean.

In practical terms, this has the consequence of increasing uncertainty around the global mean relative to a model with only a global smoother. In some cases, it can result in the estimated global smoother being close to flat, even in simulated examples with a known strong global effect. This concurvity issue may also increase the time it takes to fit these models (e.g., compare the time it takes to fit models *GI* and *I* in Table 5). These models can still be estimated because of penalty terms; all of the methods we have discussed for fitting model *GS* (factor-smoother terms or random effect tensor products) automatically create a penalty for the null space of the group-level terms, so that only the global term has its own unpenalized null space. Both the REML and ML criteria work to balance penalties between nested smooth terms (this is why nested random effects can be fitted). We have observed that **mgcv** still occasionally finds solutions with simulated data where the global term is over-smoothed.

To avoid this issue, we recommend both careful choice of basis and relative degrees of freedom of global and group-level terms. In the examples in section "What are Hierarchical GAMs?," we used smoothers with an unpenalized null space for the global smoother and ones with no null space for the group-level terms. When using TPRS this can be achieved with splines with a lower order of derivative penalized in the group-level smoothers than the global smoothers, as lower-order TPRS have fewer basis functions in the null space. For example, we used `m=2` (penalizing squared second derivatives) for the global smoother, and `m=1` (penalizing squared first derivatives) for group-level smoothers in models *GS* and *GI*. Another option is to specify the maximum degrees of freedom (`k`) for the group-level smoother either substantially higher or lower than the global smoother; this is in effect an approximate way to specify a prior belief in the relative smoothness of the global vs. group-level functions. If the group-level term is set to have greater `k` compared to the global term, this encodes the assumption that the global function should not be very wiggly, but group-level deviations from that smooth might vary from that, and vice versa if `k` is set lower for group-level terms than for the global smoother.

As noted above, interpreting the shape of global terms and group-wise deviations separately for *GS* models fit using tensor-product group-level terms is complicated by the fact that **mgcv** will drop some basis-functions from the group-level terms to prevent

[10] There is an important caveat here. When fitting *GS* models using tensor products in **mgcv**, the global and group-level terms will not be entirely concurve because **mgcv** will automatically drop basis functions from the group-level smoother to ensure that these terms are not perfectly concurve. That is, so that no basis function in the global term could be formed from a linear combination of group-level basis functions (see `?mgcv::gam.side` for how terms to be dropped are selected). Group-level terms fit using `bs="fs"` smoothers will not have any basis functions dropped, as **mgcv** disables checking for side-constraints for these smoothers (since all basis functions are fully penalized for this type of smoother, in principle concurvity should not be an issue; see `?mgcv::smooth.construct.fs.smooth.spec` for details).

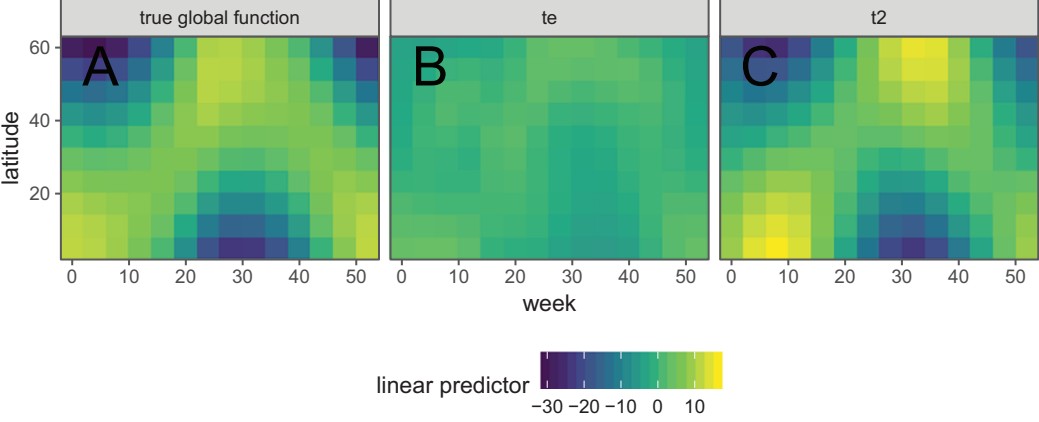

**Figure 20 Average global function used for simulating `bird_move` data set (A) compared to the fitted global function for a *GS* model estimated with either a `te()` smoother (B) or a `t2()` smoother with `full=TRUE` (C) for group-level terms.** Both group-level smoothers used the same model specification as in section "What are Hierarchical GAMs?" except for the type of tensor product used. Colors indicate the value of the linear predictor of bird density at each location in each week.

[11] It is also important to consider here that the concept of a "global function" is a bit fuzzy itself, and there are many possible ways to define what a global function is (as we discussed in section "What are Hierarchical GAMs?"). The global function being fit in all of these models is actually an average function, and the shape of it will depend on the sampling structure of any given study. In our view, the global function fitted in these models should generally be viewed as a useful summary of an average trend across a wide range of groups, and would only represent an actual average relationship if the grouping levels were drawn at random from some underlying population and if there was scientific reason to believe that individual groups should differ from the mean only via some additive function.

perfect concurvity. For tensor-product smoothers, **mgcv** will generally drop $\leq K$ terms from the group-level smoother, where $K$ is the number of basis functions in the global smoother. The total number of terms dropped will depend on the smoothers used for the global and group-level terms. This means that some groups will have a different range of potential deviations from the global smoother than others. This has the effect of also somewhat altering the shape of the global smooth relative to what it would be based on model *G* (the average curve through all the data); this will be a larger issue when the number of basis functions in the global smooth and the number of group levels are small. We have tested the effect of this issue on our simulated `bird_move` data set and did not find that it lead to substantial bias in estimating the shape of the global smoother, relative to the amount of bias inherent in any smooth estimation method[11] (Fig. 20). As noted in the section "What are hierarchical GAMs?", we found that `t2()` tensor product smoothers with full penalties (`full = TRUE` in the `t2()` function) for group-level smoothers showed the best performance at recreating the true global function from our simulated `bird_move` data set, compared to other possible types of tensor product. Using `te()` tensor products for the group-level terms lead to the global smoother being heavily smoothed relative to the actual average function, used to simulate the data (Fig. 20). However, more work on when these models accurately reconstruct global effects is still needed.

There is currently no way to disable dropping side constraints for these terms in **mgcv**. In cases where accurately estimating the global smoother or group-level deviations is essential, we recommend either fitting model *G*, *GS* using factor-smooth group-level terms (`bs="fs"`, which can also be used to model multi-dimensional isotropic group-level smoothers), or model *GI*. Alternatively, there is specialized functional regression software such as the `pffr` function in the **refund** package (*Scheipl, Gertheiss & Greven, 2016*), which does not impose these side constraints; instead the package uses a modified type of tensor-product to ensure that group-level terms sum to zero at each level of the

predictor (*Scheipl, Gertheiss & Greven, 2016*). See below for more information on functional regression.

## A brief foray into the land of Bayes

As mentioned in the section "A Review of Generalized Additive Models," the penalty matrix can be interpreted as the prior precision (inverse prior covariance) matrix for the model parameters β. Intuitively, the basis functions and penalty are an informal prior on how we'd like our model term to behave. REML gives an empirical Bayes estimate of the smooth model (*Laird & Ware, 1982*), where terms in the null space of the smoother have improper, flat priors (i.e., any value for these terms are considered equally likely). Any terms in the range space are treated as having a multivariate normal prior, and the smoothing parameters are treated as having an improper flat prior (see *Wood (2017a)*, Section 5.8 for more details on this connection). The posterior covariance matrix (*Wood, 2006b*) for model parameters can be extracted from any fitted `gam()` or `bam()` model with `vcov (model)` (this is conditional on the estimated smoothing parameters unless the option `unconditional=TRUE` is specified). Given the normal posterior for the estimates of β, we can sample from a multivariate normal with mean $\hat{\beta}$ and posterior covariance matrix. Such samples can be used to estimate uncertainty in functions of the predictors. Viewing our GAM as Bayesian is a somewhat unavoidable consequence of the equivalence of random effects and splines—if we think that there is some true smoother that we wish to estimate, we must take a Bayesian view of our random effects (splines) as we do not think that the true smoother changes each time we collect data (*Wood, 2017a*, Section 5.8).
The standard confidence intervals used in **mgcv** are in fact Bayesian posterior credible intervals, which happen to have good frequentist across-the-function properties (*Wood, 2006b*; *Marra & Wood, 2012*). The newest version of **mgcv** as of this writing (v. 1.8–28) also includes an experimental implementation of integrated nested Laplace approximation to calculate full posterior distributions for GAMs, via the `ginla` function (*Wood, in press*).

This also means that HGAMs can be included as components in a more complex fully Bayesian model. The **mgcv** package includes a function `jagam()` that can take a specified model formula and automatically convert it into code for the JAGS (or BUGS) Bayesian statistical packages, which can be adapted by the user to their own needs.

Similarly, the **brms** package (*Bürkner, 2017*), which can fit complex statistical models using the Bayesian software **Stan** (*Carpenter et al., 2017*) allows for the inclusion of **mgcv**-style smooth terms as part of the model specification. The **brms** package does not currently support `te()` tensor products, but does support factor-smooth interactions and `t2()`-style tensor products, which means all of the models fitted in this paper can be fitted by **brms**. Finally, the **bamlss** package (*Umlauf, Klein & Zeileis, 2018*) can fit complex GAMs using a number of computational backends, including JAGS and BayesX, using **mgcv** syntax for model specification.

## Beyond HGAMs: functional regression

The HGAMs we have discussed are actually a type of *functional regression*, which is an extension of standard regression models to cases where the outcome variable $y_i$ and/or the

predictor variables $x_i$ for a given outcome are functions, rather than single variables (*Ramsay & Silverman, 2005*). HGAMs as we have described them are a form of function-on-scalar regression (*Ramsay & Silverman, 2005*; *Reiss, Huang & Mennes, 2010*), where we are trying to estimate a smooth function that varies between grouping levels. Here the "scalar" refers to the grouping level, and the function is the smooth term that varies between levels; in contrast, a standard GAM is a type of scalar-on-scalar regression, as the goal is to use a set of single values (scalars) to estimate each (scalar) response.

We have deliberately focused our paper on these simpler classes of functional regression model, and chosen to use the term HGAM rather than functional regression, as we believe that this more clearly connects these models to modeling approaches already familiar to ecologists. Further, we consider the unit of analysis to still be individual observations, as compared to functional regression where the unit of analysis is whole functions. For instance, we are interested in applications such as species distribution modeling, where the presence of a given species may be predicted from a sum of several species-specific functions of different environmental variables.

However, there is an extensive literature dedicated to the estimation of more complex functional regression models for any interested reader (see *Morris (2015)* and *Greven & Scheipl (2017)* for a good introduction and overview of more recent work in this field). The **refund** package (*Greven & Scheipl, 2017*) uses the statistical machinery of **mgcv** to fit these models, and should be usable by anyone familiar with **mgcv** modeling syntax. Functional regression is also a major area of study in Bayesian statistics (*Kaufman & Sain, 2010*).

## CONCLUSION

Hierarchical GAMs are a powerful tool to model intergroup variability, and we have attempted to illustrate some of the range and possibilities that these models are capable of, how to fit them, and some issues that may arise during model fitting and testing. Specifying these models and techniques for fitting them are active areas statistical research, so this paper should be viewed as a jumping-off point for these models, rather than an end-point; we refer the reader to the rich literature on GAMs (*Wood, 2017a*) and functional regression (*Ramsay & Silverman, 2005*; *Kaufman & Sain, 2010*; *Scheipl, Staicu & Greven, 2014*) for more on these ideas.

## ACKNOWLEDGEMENTS

The authors would like to thank Carly Ziter, Tiago Marques, Jake Walsh, Geoff Evans, Paul Regular, Laura Wheeland, and Isabella Ghement for their thoughtful feedback on earlier versions of this manuscript, and the Ecological Society of America for hosting the **mgcv** workshops that this work started from. The authors also thank the three reviewers (Paul Bürkner, Fabian Scheipl, and Matteo Fasiolo) for their insightful and useful feedback.

All authors contributed to developing the initial idea for this paper, and to writing and editing the manuscript. Author order after the first author was chosen using the code:

```
set.seed(11)
sample(c('Miller','Ross','Simpson'))
```

All code used to generate this paper, as well as prior versions of this manuscript, are available at: github.com/eric-pedersen/mixed-effect-gams.

### Funding

This work was funded by Fisheries and Oceans Canada, Natural Science and Engineering Research Council of Canada (NSERC) Discovery Grant (RGPIN-2014-04032), by OPNAV N45 and the SURTASS LFA Settlement Agreement, managed by the U.S. Navy's Living Marine Resources Program under Contract No. N39430-17-C-1982, and by the USAID PREDICT-2 Program. There was no additional external funding received for this study. The funders had no role in study design, data collection and analysis, decision to publish, or preparation of the manuscript.

### Grant Disclosures

The following grant information was disclosed by the authors:
Fisheries and Oceans Canada, Natural Science and Engineering Research Council of Canada (NSERC) Discovery Grant: RGPIN-2014-04032.
OPNAV N45 and the SURTASS LFA Settlement Agreement, managed by the U.S. Navy's Living Marine Resources program: N39430-17-C-1982.
USAID PREDICT-2 Program.

### Competing Interests

Eric Pedersen is an employee of Fisheries and Oceans Canada, and Noam Ross is employed by EcoHealth Alliance, a nonprofit organization.

### Author Contributions

- Eric J. Pedersen conceived and designed the experiments, performed the experiments, analyzed the data, prepared figures and/or tables, authored or reviewed drafts of the paper, approved the final draft.
- David L. Miller conceived and designed the experiments, prepared figures and/or tables, authored or reviewed drafts of the paper, approved the final draft.
- Gavin L. Simpson conceived and designed the experiments, prepared figures and/or tables, authored or reviewed drafts of the paper, approved the final draft.
- Noam Ross conceived and designed the experiments, prepared figures and/or tables, authored or reviewed drafts of the paper, approved the final draft.

### Data Availability

All data and code for this article are available via GitHub:
https://github.com/eric-pedersen/mixed-effect-gams

Eric J. Pedersen, David L. Miller, Gavin L. Simpson, & Noam Ross. (2019, February 25). *Hierarchical Generalized Additive Models: an introduction with mgcv (Version v0.2.0).* Zenodo. DOI 10.5281/zenodo.2577381.

## Supplemental Information

Supplemental information for this article can be found online at http://dx.doi.org/10.7717/peerj.6876#supplemental-information.

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
