# Peer review of "Hierarchical generalized additive models in ecology: an introduction with mgcv"

_PeerJ, doi:10.7717/peerj.6876_

## Round 0.1 · original submission · Minor Revisions

First of all, I would like to greatly thank the reviewers for their insightful and detailed comments. I don’t think that any of their comments will require major revisions and all deserve a response. I will ask you to respond to each comment in turn and either indicate changes to the manuscript in response to that comment or explain why you have not made any such changes.

While I agree with Reviewer 3’s comment about adding “ecology” to the title, I also think that this manuscript would be of considerable interest to a wide range of readers (and is the type of manuscript I would greatly enjoy presenting at a journal club to a mixed audience). I look forward to seeing your revised version of the manuscript.

A small number of minor typos/comments that I don’t think our eagle-eyed reviewers spotted and mentioned:

Line 149: “derivatives” not “derivative”

Line 202: I’d qualify this with “in ecological” as common groupings vary between disciplines and I think this manuscript should be of interest to many outside of the ecological community

Line 304: Missing space after comma following “instance”

Line 369.5: Spurious backtick

Lines 433, 436, 438, 449, 543, 545, 560, 564, 662, 670, 679, 681, and 755: Missing reference (Reviewers 1 and 3 noted these also, I’m just adding the full set of line numbers here).

·

Basic reporting

see below

Experimental design

see below

Validity of the findings

see below

Additional comments

This is a strong paper and I only have a few minor comments.

1. Line 34: HGLMs may not only handle linear but also quadratic etc. relationships.
2. Line 37: Did you mean something like “may vary across groups”?
3. Line 182: typo in “average average”
4. In some Figures (e.g. Figure 7), points are hidden by the prediction line. Can this be changed?
5. Line 408: It should be made clear that the listed smoothers are the remaining ones not those being removed.
6. A lot of Tables have broken links as indicated by “Table ??”.
7. Line 438: It seems as if model 4 is slightly better than model 2 contrary to what the text says.
8. Line 489: typo in “could could”
9. Line 501: What’s a “model-to-model comparison” as compared to a “model comparison”?
10. Line 509: Why are fitted vs. response plots less useful for non-gaussian families?
11. Line 520 ff.: Could this be made a proper table with some notes? Also, there seem to be spaces in the category names in the table. I think it would be better to remove these spaces.
12. Figure 14: It should be made explicit to which data this plot belongs. Also, it seems to be left and right not “Top” and “bottom” row.
13. Figure 15: Maybe the y-axis should include explicitly that predictions are on the log-scale (not just in the note).
14. Table 3: I think it’s a little bit concerning that the intercept only has not much worse (or even better) predictions despite a clear non-linear trend. Perhaps it’s worth discussing that a little bit more thoroughly.
15. Line 619 (and all lines where you compare models based on AIC): On which basis do you decide when to models differ “substantially” in their AIC values?
16. Line 657: Ist n_g and g ment to be the same? At least g is not introduced before it is used.
17. Line 787: You say that gam() uses the Bayesian posterior covariance matrix. As gam() is frequentist I would naively expect this covariance matrix to be an *approximation* of the Bayesian posterior covariance matrix, or is it really an exact one? In any case, it can only be the mode of this covariance matrix unless we obtained the full posterior distribution somehow.
18. Line 804: With the latest CRAN releases of mgcv and brms, factor-smooth interactions are now possible.
19. Line 810: It may be helpful to describe in a little bit more detail how GAMs fit into the functional regression framework. For instance, in the term “function-on-scalar” it may be unclear for readers if x or y is the function / the scalar.


Comments regarding the R code:

1. I think more spaces within lines of code as well more new-lines would help with the readability. I would recommend single lines not exceeding 80 characters.
2. Longer comments are perhaps better put into their own lines
3. It is saver not to abbreviate TRUE and FALSE by T and F respectively.
4. Personally, I like the style
object <- fun(
<arguments>
)
in order to prevent arguments from starting at the 40+ character in the line, but it’s just a matter of style that I don’t want to impose on anyone.
5. Perhaps it would be beneficial to the readability if you used an Rmarkdown document instead of an R script to have comments outside of code chunks. As a bonus you could compile the document to html so that people can directly look at the results next to the code without having to run the code themselves.

·

Basic reporting

No major comments, this is mostly very well written and structured.

Experimental design

The simulation study re. over/undersmoothing in Section V needs more work, in my opinion, or at least a much more circumspect interpretation of the result(s).

Validity of the findings

See above re. Section V, see attached for specific small points on model checking and representation of results.

Additional comments

See attached. Some typos, some snetences could be more clear, some minor aspects of results I could not reproduce or interpretations I did not share.

·

Basic reporting

The article provides an introduction to Hierarchical GAMs in mgcv, which will be useful to scientists interested in analysing data sets with a hierarchical structure. The article is generally clear, and it provides the right level of detail for the intended audience. I have only major comment about the presentation:

- In the paper you frequently refer to models 1 to 5, so that a forgetful reader is forced to frequently go back to their definition in Figure 4. I think that it would be better to use more meaningful labels for the models, such as:
1) G (*G*lobal only)
2) GIC (*G*lobal + *I*ndividual effects with *C*ommon penalty)
3) GIM (*G*lobal + *I*ndividual effects with *M*ultiple penalties)
4) IC (only *I*ndividual effects with *C*ommon penalty)
5) IM (only *I*ndividual effects with *M*ultiple penalty)
Obviously the labels above are just a suggestion, but the current 1 to 5 labels are making the discussion in the paper quite hard to follow.

Experimental design

No comment.

Validity of the findings

The examples used in the paper are clear and should be useful to any applied scientist wanting to start using HGAM in mgcv. I have two main comments.

1) When analysing the D. mendota data set, the author assess the four model considered using the RMSE on the test set. Table 3 shows that the GAM models do slightly better than the intercept-only model on the first 3 data sets and slightly worse on the Waubesa data. However, the Waubesa data contains a very large outlier, which has a strong influence on the RMSEs. Using the Mean Absolute Error leads to different results:

# A tibble: 4 x 5
Lake `Intercept only` `Model 1` `Model 2` `Model 3`
<fct> <chr> <chr> <chr> <chr>
1 Kegonsa 1.7 1.5 1.5 1.4
2 Mendota 2 1.6 1.6 1.6
3 Menona 1.9 1.6 1.6 1.6
4 Waubesa 2.6 2.5 2.7 2.6

My point is that the reader might look at table 3, and conclude that it is not worth spending much time learning about HGAMs, given that their performance is very close to that of an intercept-only model on real data. To avoid this, it would be important to point out that table 3 (and probably table 2) are based on relatively small data sets with occasional very large (and unpredictable) responses. Hence it is difficult to see how (in the absence of additional information) any statistical model could lead to a much lower RMSE on the test set, relative to the intercept-only model.

2) Looking at Figure 18, one might get the impression that the bam() function is pretty useless, given that it is beaten by gamm and gamm4 for any group size. However, it would be important to point out that bam scales well (possibly better than any other fitting methods in mgcv) with the number of observations per group, especially if the discrete = TRUE option is used. It might also be useful to include the results of a comparison where the number of groups is held constant, and the number of observations within each group is increased.

Additional comments

I have some relatively minor comments and suggestions:

- Given that the paper is aimed specifically at ecologists, maybe the word "ecology" should appear already in the title.

L 105: "A penalty term is then added to the model likelihood", this is the log-likelihood, right?

L 109: "In the left plot λ was estimated" probably I am being pedantic but I would say that lambda is
being "selected" not "estimated".

L 150: Here you are talking about the penalty used in TPRS. Maybe it would be useful to add a
pointer to the relevant section in Wood's book or to add a formula for the penalty, which
is described a bit vaguely at the moment.

Caption to figure 3: "This means these functions are in the null space of this basis" the intercept and
linear term are in the null space of the penalty matrix, not of the basis. Of course
the smooth could then be re-parametrized to separate null space and span,
but I don't think you are mentioning the re-parametrization here.

L 364 "This increases the computational cost of the model (as there are more smoothing parameters to
estimate), and means that the only information shared between groups is through the global
smoother." Notice that information is shared also for estimating the variance of the
observations, and for estimating the variance of the random effects (if these are included as in
the formula following line 369).

L 369 I am a bit confused about what is happening here:
y ~ s(x, bs="tp") + s(x, by=fac, m=1, bs="ts") + s(fac, bs="re")`
In particular, the by-factor effects have a first order penalty hence their null space should be
empty if these smooths have been centred (that is, if the constant function has been removed
from their span). If this is correct, then what is gained by using "ts" rather than "tp"?
Please clarify (I have a similar doubts about the plants and birds models).

L 379 "Using ts helps reduce this issue, as the only unpenalized null space terms will occur in
the global smoother." maybe it would be clearer to say "Using ts helps reduce this issue,
as the unpenalized null space terms will occur only in the global smoother."
(I moved the "only" around).

L 417 "count ~ t2(week, latitude, species, bs=c("cc", "tp", "re"), k=c(10, 10, 6), m=c(2, 2, 2))"
What is the meaning of using a 2nd order penalty for the random effect here? I guess that
you had to set m to c(2, 2, 2) rather than c(2, 2) because you need m to have the same
length as the number of marginals, but the last 2 will not be used.
Please clarify this in the text as a non-expert reader might get confused.

L 420 For bird_mod5 you are still using bs = "ts", but there is no identifiability issue here right?
Please clarify or change to "tp".

L 516 "The k.check test shows that the default maximum degrees of freedom for the smoothers used in model 5 are sufficient for all species, as the p-value for the observed k-index" personally I prefer looking at whether EDF is close to k', rather than looking at the p-values. The reason is that the p-values can be highly variable under the null hypothesis (no residual autocorrelation), and can change from (say) 0.4 to 0.02 from one call to gam.check to the next. Further if one of the p-values is, say, 0.01 while edf=2 and k'=10 I don't see why one should consider increasing k further for this effect (while if edf=9.8, k'=10 I would consider increasing k even if the corresponding p-value was not significant).

L 537 "Note that there is no p-value for the random effects smoothers s(taxon) and s(taxon,year_f) as the k value for this term is already set at its maximum." It would be worth pointing out that the p-values come from simulation-based tests for autocorrelation of the residuals. So this test cannot be performed for factors such as taxon (there is no natural way of ordering the residuals along taxon).

Table 3: "Table 3: Out-of-sample predictive ability for model 1-3 applied to the D. mendotae dataset. RMSE values represent the average squared difference between model predictions and observations for held-out data (zero predictive ability would correspond to a RMSE of one)" Are you sure that the RMSE should be 1 for a very bad model?

L 583 "This estimate would have no variance (as it is always the same regardless of what the data look like) but would have high bias unless the true population mean happened to equal zero." should be "... unless the true population mean happened to equal to the predetermined fixed value for the mean." (I think you are mixing what you write in footnote 6 with the main text here)

L 650 "(i) the number of basis functions to be estimated" strictly speaking the basis function are fixed, their regression coefficients have to be estimated

L 849 "All code used to generate this paper, as well as prior versions of this manuscript, are available
at: github.com/noamross/mixed-effect-gams." I would include this sentence also at the end of the abstract, so that the reader will know immediately where the code is.


A list of typos:

L 144 " can be use for problems"

L 174 "one wanted estimate the interacting"

L 227 " but very different shapes. wiggliness measures"

In footnote 1 on page 9: "it will will raise" and also "error message if passed data coded as character"

L 433 "Table ??. " Notice that the references to all tables are broken.

L 508 "The code for model 4 and 5 for the community zooplankton model is shown here:" but the formulas actually are at top of the page.

L 670 "need to be estimate"

L 686 "the funcion bam()"

Figure 18: Notice that the legend does not allow to distinguish between bam(discete = TRUE) and bam(discrete = FALSE)

L 846 "the intial idea"

---

## Round 0.2 · Minor Revisions

Thank you for your thoughtful revisions. The first and third reviewers had no comments on your revised version and the second reviewer only had three very minor comments that I do not think will require much effort on your part to address. Their first comment seems to be an issue only for the diff version, with the code evident in the revised manuscript (between Lines 537 and 538).

·

Basic reporting

no comment

Experimental design

no comment

Validity of the findings

no comment

Additional comments

The authors have addressed all of my previous comments and I have no further suggestions.

·

Basic reporting

All issues previously reported have been answered.

(Very) minor points (NB: page and line numbers refer to pdf with diffs):

p. 27, l. 578:
"The code for model 4 and 5 checking model S and I for the community zooplankton model is:"
<Code not actually shown>

p. 43 bottom/44 top:
- please refer to function "refund::pffr" not "fosr" - "fosr" is fairly limited and does not use mgcv::gam.
- please cite "Scheipl, Fabian; Gertheiss, Jan; Greven, Sonja. Generalized functional additive mixed models. lectron. J. Statist. 10 (2016), no. 1, 1455--1492. doi:10.1214/16-EJS1145. https://projecteuclid.org/euclid.ejs/1464710238"; here, which is the more relevant reference for this instead of Scheipl/Staicu/Greven 2014.

p. 44, l. 985f.:
Maybe worth mentioning here that, starting from the current version, mgcv now does approximate fully Bayesian inference via a clever version of INLA as well (mgcv::ginla).
None of us could ever hope to keep up with Simon Wood...

Experimental design

All issues previously reported have been answered thoroughly.

Validity of the findings

All issues previously reported have been answered thoroughly.

Additional comments

It was a pleasure to read the author's well-written and thoughtful rebuttal and the revised and much improved manuscript.
Congratulations on a very interesting and insightful paper.

·

Basic reporting

NA

Experimental design

NA

Validity of the findings

NA

Additional comments

I am happy with the revised version of the manuscript, and with the authors' reply to my comments. I have no further suggestions.

---

## Round 0.3 · accepted · Accept

It is my pleasure to accept your manuscript with those final few edits. Well done on writing what I think will be a very useful article to researchers looking at using GAMs with clustered or longitudinal data, for those working both within and outside ecology.

#